# Mechanistic characterization of a *Drosophila* model of paraneoplastic nephrotic syndrome

Jun Xu [1,2,4] ✉, Ying Liu [2,4] ✉, Fangying Yang[1], Yurou Cao[1], Weihang Chen[2], Joshua Shing Shun Li[2], Shuai Zhang[1], Aram Comjean [2], Yanhui Hu [2] & Norbert Perrimon [2,3] ✉

Paraneoplastic syndromes occur in cancer patients and originate from dysfunction of organs at a distance from the tumor or its metastasis. A wide range of organs can be affected in paraneoplastic syndromes; however, the pathological mechanisms by which tumors influence host organs are poorly understood. Recent studies in the fly uncovered that tumor secreted factors target host organs, leading to pathological effects. In this study, using a *Drosophila* gut tumor model, we characterize a mechanism of tumor-induced kidney dysfunction. Specifically, we find that Pvf1, a PDGF/VEGF signaling ligand, secreted by gut tumors activates the PvR/JNK/Jra signaling pathway in the principal cells of the kidney, leading to mis-expression of renal genes and paraneoplastic renal syndrome-like phenotypes. Our study describes an important mechanism by which gut tumors perturb the function of the kidney, which might be of clinical relevance for the treatment of paraneoplastic syndromes.

Paraneoplastic syndromes are a group of disorders associated with tumors that are prevalent in patients with lymphatic, lung, ovarian or breast cancers[1,2]. These syndromes are not attributable to tumor invasion or compression but are due to the response of body systems and organs to tumors and/or the factors that they secrete[3,4]. As such, characterization of tumor-host interactions might help reveal mechanisms underlying paraneoplastic syndromes. However, due to the complexity of tumor-host interactions in patients and in mammalian cancer models, studies in alternative and simpler models of the mechanisms underlying paraneoplastic syndrome pathogenesis are needed.

A wide range of organs can be affected by cancer, and dysfunction in different body systems leads to distinct paraneoplastic syndromes[3]. For instance, dysfunctions in the nervous system caused remotely by tumors, termed paraneoplastic neurologic disorders, lead to a diverse group of symptoms such as encephalitis, optic neuropathy, and cerebellar degeneration[5]. In other cases, polymyositis and hypertrophic osteoarthropathy have been reported as paraneoplastic syndromes caused by influence of lung cancer on muscle and bone, respectively[6,7]. Defects in the kidney have also been observed in patients with cancer. However, in these instances, it is not clear whether the renal dysfunctions are caused by treatment of the cancer or whether tumor-secreted factors such as hormones and cytokines are the primary cause[8–10].

*Drosophila* has emerged as a simple and powerful model to identity pathogenic tumor-host interactions[11,12]. The fly has organs that are functionally equivalent to human organs, including digestive tract, liver, adipose tissue, and kidney[13], and a number of models have been established to study organ-specific disorders triggered by tumors[11,12]. For instance, elevated JAK-STAT signaling in a fly tumor model disrupts the blood-brain barrier leading to death[14]. In addition, two tumor-secreted factors, Unpaired 3 (Upd3) and PDGF- and VEGF-related

[1]CAS Key Laboratory of Insect Developmental and Evolutionary Biology, CAS Center for Excellence in Molecular Plant Sciences, Shanghai Institute of Plant Physiology and Ecology, Chinese Academy of Sciences, Shanghai, China. [2]Department of Genetics, Blavatnik Institute, Harvard Medical School, Boston, MA, USA. [3]Howard Hughes Medical Institute, Boston, MA, USA. [4]These authors contributed equally: Jun Xu, Ying Liu. ✉e-mail: junxu@cemps.ac.cn; ying_liu@hms.harvard.edu; perrimon@genetics.med.harvard.edu

factor 1 (Pvf1), target host muscle and fat body to stimulate body wasting, which is reminiscent of cachexia[15,16]. Here, leveraging a *Drosophila* gut tumor model, we demonstrate that tumor-secreted PDGF- and VEGF-related factor 1 (Pvf1) stimulates PDGF/VEGF signaling in the fly malpighian tubules (MTs), a renal system functionally equivalent to the vertebrate kidney[17,18], leading to formation of kidney stones, ascites/bloating, and uric acid accumulation. Importantly, these renal phenotypes contribute to increased mortality of flies with gut tumors. Altogether, our data suggests that tumors remotely induce renal dysfunction through hijacking of normal signaling pathways, which represents an underappreciated effect of paraneoplastic renal syndromes. Our findings that a tumor can induce renal disorders, independent of cancer treatment, might open new therapeutic avenues to treat kidney-related symptoms in patients with cancer.

## Results

### Kidney dysfunction contributes to bloating and reduces lifespan

Flies with gut tumors induced by activated *yorkie* (esg> *yki^{act}*; referred to hereafter as Yki flies) exhibit a 'bloating phenotype' characterized by an enlarged, fluid-filled abdomen[19]. This suggests the possibility of a defect in the fly renal system, the Malpighian tubules (MTs), which maintain fluid and electrolyte balance in the fly hemolymph[20]. Interestingly, the MTs of Yki flies contain crystals (kidney stones) in their lower segments, which increase in size upon tumor progression (Fig. 1A). The accumulation of kidney stones can be driven by many factors, including elevated levels of uric acid[21,22]. Indeed, uric acid levels are increased in Yki flies as compared to controls (Fig. 1B). To assess the role of uric acid levels in the formation of kidney stones, we fed Yki flies with allopurinol, which can repress uric acid levels by inhibiting purine degradation[21]. Although wildtype flies are tolerant, this drug led to severe lethality of Yki flies. As an alternative strategy, we increased uric acid levels by feeding Yki flies with high purine food, which induced kidney stones in the main segment of the MTs (Fig. 1C, D), indicating that uric acid levels positively correlate with kidney stone formation. To determine if formation of kidney stones impacts renal function, we fed Yki flies with 3% *Garcinia cambogia* extract that is known to dissolve kidney stones in flies[23], and observed a reduction in average kidney stone sizes from 18.12 μm to 4.67 μm without affecting the gut tumors (Fig. 1E). Interestingly, *Garcinia cambogia* feeding significantly decreased the prevalence of bloating among tumor flies from 86.4% to 35.5% (Fig. 1F, G). Consistent with this, the average wet body weight of tumor flies was reduced from 2.52 mg to 1.54 mg, while the dry mass remained unchanged (Fig. 1H). Yki flies fed with *Garcinia cambogia* did not show a reduction in uric acid levels (Supplementary Fig. 1), likely because hydroxycitric acid dissolves kidney stones by inhibiting calcium oxalate monohydrate nucleation[24]. Next, to determine whether kidney stones lead to reduced lifespan, we fed flies with 0.5% calcium oxalate (NaOx), which induces stone formation in the lumen of MTs[25]. NaOx-fed flies showed an increase in kidney stone size and reduced lifespan (50% mortality was shifted from 18 days to 5 days) (Fig. 1I, J). Altogether, these data suggest that kidney disfunction contributes to the imbalance of body fluids and mortality of Yki flies.

### Single-cell survey reveals abnormal cell composition of MT

To gain a comprehensive understanding of renal dysfunction in Yki flies, we performed a single nuclei RNA-sequencing (snRNA-seq) analysis of MTs from control and Yki flies. We analyzed MTs at day 8 upon tumor induction when the bloating phenotype is apparent[15]. In total, 1706 and 1849 cells from control and Yki flies, respectively, were recovered from snRNA-seq. Cell clusters were annotated according to our previous report of marker genes in MTs (Fig. 2A)[26]. Nine cell clusters were identified, including initial and transitional principal cells (PCs), main segment PCs, lower tubule PCs, upper ureter PCs, lower ureter PCs, lower segment PCs, renal stem cells (RSCs), and stellate cells (SCs) (Fig. 2A). An additional cluster was identified in the Yki sample, which we labeled as 'main segment PCs 2' as they express *List* (Fig. 2A), a previously identified main segment PC marker gene[26]. The top two marker genes for the 'main segment PCs 2' cluster are *Slc45-1* and *CAHbeta*, an osmolyte/sucrose transporter and a beta-class carbonate dehydratase, respectively[27,28]. To assess changes in MTs in response to Yki gut tumors, we analyzed the respective number of cells within each cluster. Yki fly MTs displayed a decrease in main segment PCs and an increase in upper ureter PCs and lower segment PCs (Fig. 2B), suggesting abnormalities in these renal segments. Interestingly, the number of renal stem cells in Yki flies was significantly reduced from 19% to 6% of total cells (Fig. 2B). In line with this, *esg > GFP* positive cells in the renal stem cell zone were decreased after six days of tumor induction (Supplementary Fig. 2A, B). Previous studies have reported that Notch (N) signaling is involved in kidney stem cell renewal and maintenance (Supplementary Fig. 2C, D)[26,29]. leading us to investigate *N* expression. Expression of *N* was decreased in *yki^{act}* gut tumor flies (Log2 folder change, 0.18) compared to wildtype flies (0.58) (Supplementary Fig. 2E). In addition, downstream genes of N, *escargot* (*esg*) and *Delta* (*Dl*), were also significantly decreased in the *yki^{act}* gut tumor flies (Supplementary Fig. 2F, G). Abnormal expression levels of these genes may contribute to the renal stem cell loss in tumor flies. Finally, the *cut* (*ct*) gene, which is a marker of the principal cells, also showed decreased expression in the upper ureter principal cells and lower segment principal cells (Supplementary Fig. 2H). Taking together, these observations suggest that renal stem cell renewal and maintenance are impaired in Yki flies.

Next, we analyzed the expression levels of genes important for kidney function, which include genes involved in cell junction, kidney stone formation, cation transport, diuretic, aquaporin, and uric acid metabolism (Fig. 2C, D, Supplementary Fig. 3A–I). Consistent with the elevated uric acid levels in Yki flies, the expression of two uric acid pathway genes, *Uro* and *CG30016*, which promote uric acid excretion[21,26], was significantly down-regulated. Conversely, we observed an increase in expression in Yki flies of *Aldehyde oxidase 1* (*AOX1*) (Fig. 2C, D, Supplementary Fig. 3G, I), which catalyzes xanthine into uric acid[30]. In addition, kidney stone disease-related genes such as *CG31674*, *Vacuolar H + -ATPase 55kD subunit* (*Vha55*), and *Major Facilitator Superfamily Transporter 2* (*MFS2*) were all down-regulated in Yki flies (Fig. 2C, D, Supplementary Fig. 3C, I). Previous studies have shown that depletion of *Vha55* or *MFS2* in PCs can induce kidney stone formation[23,31]. As such, the reduced levels of these genes likely contribute to kidney stone formation in Yki flies. Further, to investigate the impaired water balance in Yki flies, we assessed the expression of genes involved in cation transport, diuresis, and water transport. Cation transport genes, including *MFS2* and *Prestin*, were down-regulated (Fig. 2C, D, Supplementary Fig. 3C, I). In addition, the *Secretory chloride channel* (*SecCl*) (2.44, p = 0.0054) was up-regulated, suggesting an impaired response to diuretic hormones (Supplementary Fig. 3D, I)[32]. Moreover, genes encoding Aquaporins, *Entomoglyceroporin 2* (*Eglp2*) (0.039, p = 0.00025), *Eglp4* (0.19, p = 0.00018), and *Prip* (0.24, p = 0.00097), were all down-regulated (Fig. 2C and D, Supplementary Fig. 3F and I). Changes in expression of these genes presumably reduce the ability of MTs to transport water, leading to excessive buildup of body fluids. Also, genes involved in cell junctions, including *Innexin 7* (*Inx7*) (1.95, p = 0.025), *Inx2* (1.74, p = 0.029), *Tetraspanin 2A* (*Tsp2A*) (6.80, p = 0.0024), *mesh* (12.93, p = 0.0027), *Snakeskin* (*Ssk*) (3.65, p = 0.00033) and *discs large 1* (*dlg1*) (1.66, p = 0.034) were up-regulated in Yki flies (Fig. 2C, D, Supplementary Fig. 3B and I), suggesting a dysregulation of renal structure. Altogether, these results indicate that gut Yki tumors remotely disturb expression of key renal genes, leading to impairment in renal functions (Supplementary Fig. 3H).

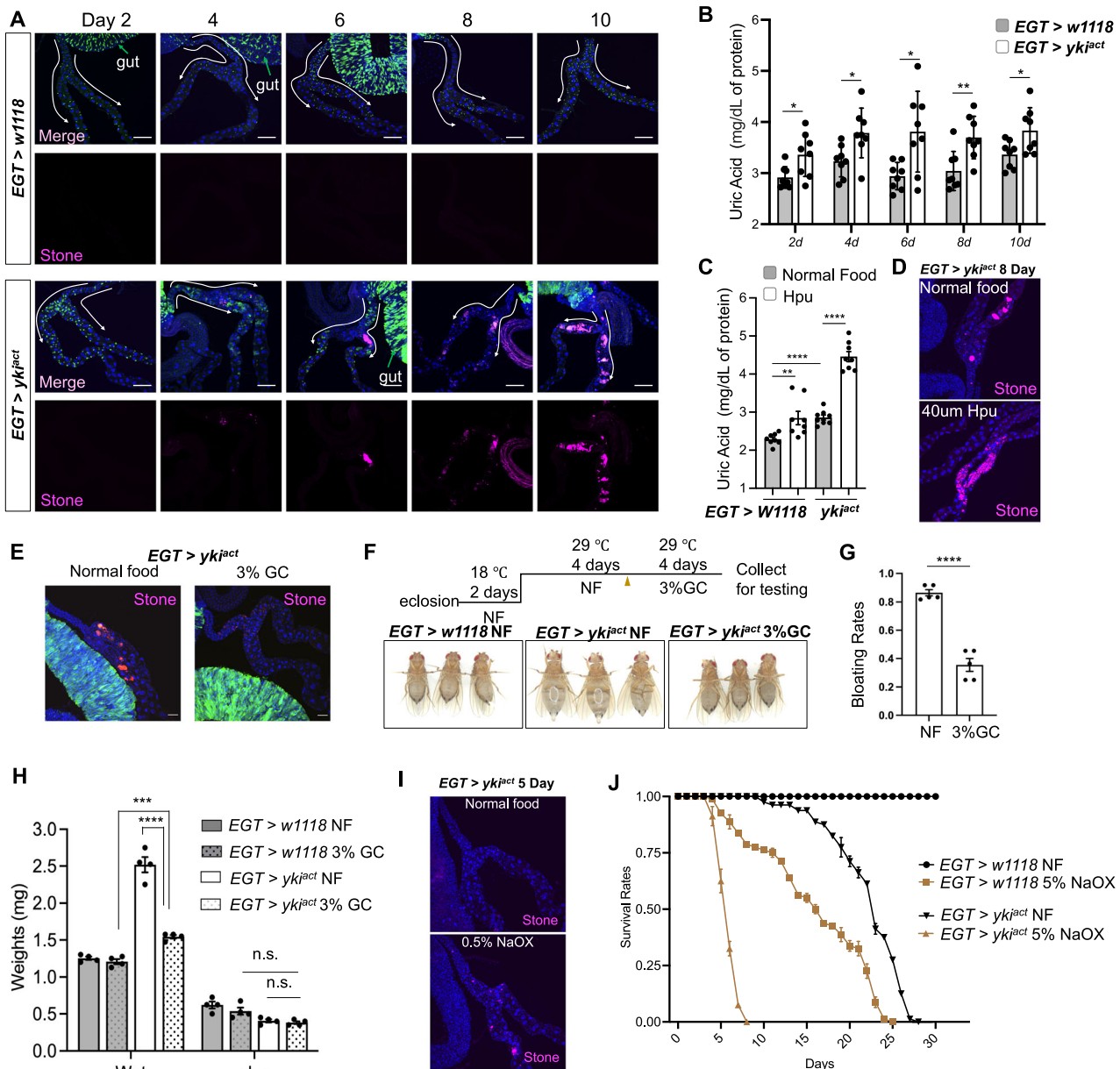

**Fig. 1 | EGT > yki^act gut tumor triggers kidney stone formation and uric acid (UA) accumulation. A** Changes in kidney stone appearance (purple) in the MT of *EGT > yki^act* flies (*esg-GAL4, UAS-GFP, tub-GAL80^TS / + ; UAS-yki^3SA/+*) and *EGT > w1118* control (*esg-GAL4, UAS-GFP, tub-GAL80^TS /+*) 2 days to 10 days after tumor induction. Green shows *esg* positive cells, blue is DAPI staining for nuclei. White arrows indicate the renal stem cell zone of MT, green arrows show the gut. **B** Whole-body UA level in *EGT > w1118* and *EGT > yki^act* flies at different time points, *n* = 8 biologically independent experiments. (C-D) High purine food (Hpu) feeding increases the amount of **C** whole body UA levels and **D** kidney stones, n = 8 biologically independent experiments in **C. E** Dissolution of kidney stones in *EGT > yki^act* flies upon feeding *Garcinia cambogia* (GC) and **F, G** inhibition of the bloating phenotype, n = 5 biologically independent experiments in **G. H** Wet/dry weights showing that GC feeding affects the water volume in the body, *n* = 4 biologically independent experiments. Feeding with sodium oxalate (NaOx) **I** increased kidney stones and **J** decreased lifespan of flies. Data are presented as means ± SEM. *p < 0.05, **p < 0.01, ***p < 0.001, ****p < 0.0001. n.s. means no significant with student t-test.

## Elevated PDGF/VEGF signaling in PCs causes kidney dysfunction

Yki tumors secrete Pvf1 and Upd3, which activate the Pvr/MEK and JAK/STAT signaling pathways, respectively, and ImpL2, which represses insulin/insulin-like signaling (IIS), in peripheral tissues[15,16,19]. To our best knowledge, none of them have been reported to influence renal function. Thus, we tested whether these secreted factors were involved in the renal dysfunction observed in Yki flies. Since genes involved in kidney stone formation and uric acid metabolism are highly enriched in PCs (Supplementary Fig. 3C, G), we first searched for Gal4 drivers that would allow us to manipulate the activity of these pathways in a PC-specific manner. The two widely used MT drivers, *C42-*

*Gal4* and *Uro-Gal4*, are either not PC-specific or are weakly expressed. Thus, we analyzed genes preferentially expressed in PCs for candidate PC-specific Gal4 drivers. Among them, *CG31272* showed strong and specific expression in all PCs, which was confirmed with a Gal4 splicing trap (CG31282[MI05026-TG4.1]) (Supplementary Fig. 4A–D). Combining this Gal4 driver with tub-Gal80ts (hereafter referred to as PC^ts), we tested activation of PDGF/VEGF and JAK/STAT signaling pathway, and inhibition of IIS in PCs of wildtype flies, mimicking the conditions in Yki flies. Only PDGF/VEGF activation (*Pvr^act*) in PCs increased whole body uric acid levels (Fig. 3A) and resulted in bloating, increased ratio of water/dry mass, and reduced lifespan, phenocopying what happens in

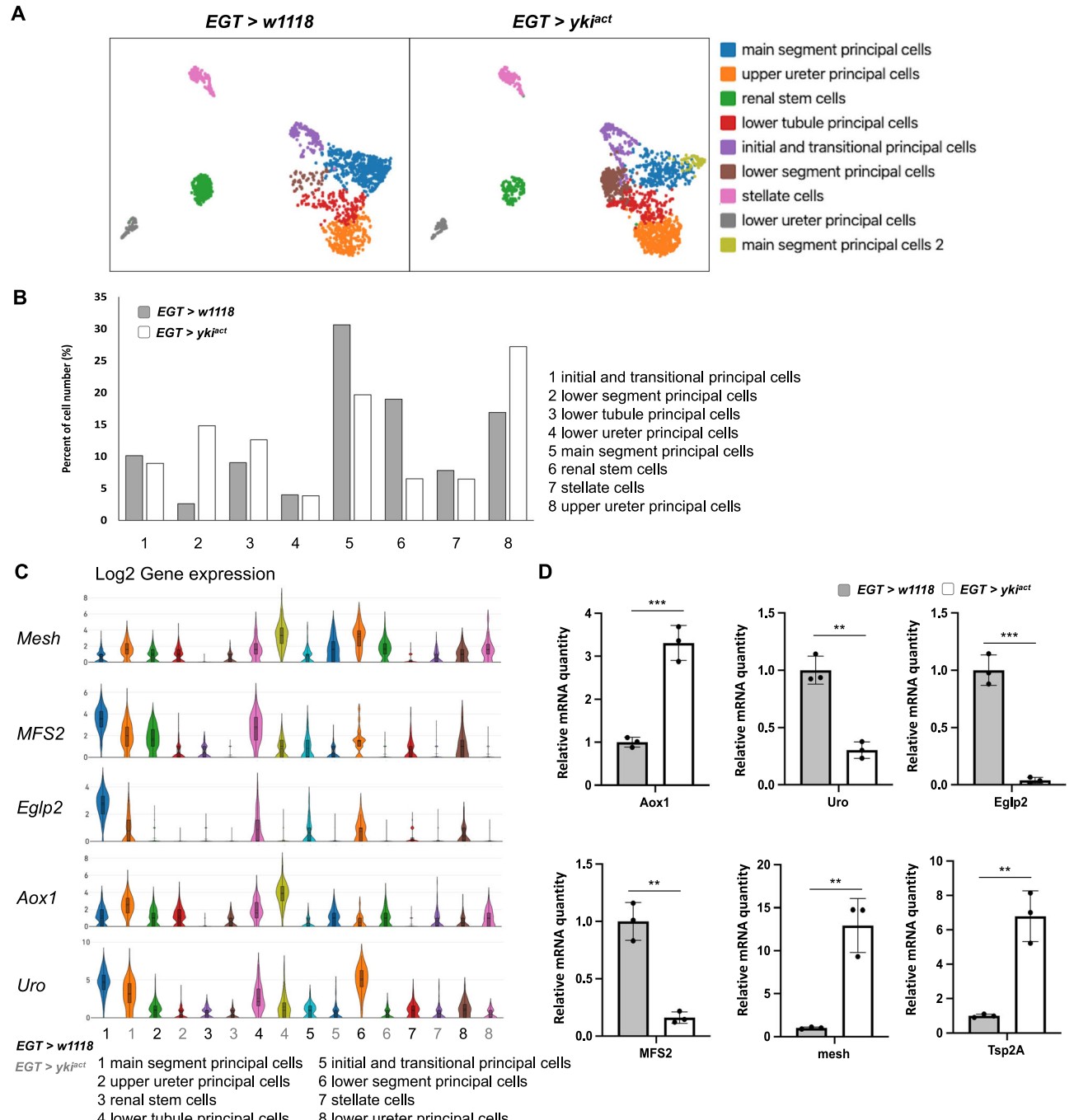

**Fig. 2 | Gene expression in MT of *EGT > yki^act* flies. A** A UMAP indicating the MT cell types of *EGT > w1118* and *EGT > yki^act* flies. Cell clusters corresponding to principal cells (initial and transitional principal cells, main segment principal cells, lower tubule principal cells, upper ureter principal cells, lower ureter principal cells, and lower segment principal cells), renal stem cells and stellate cells. **B** Percentage of cells in each cluster in *EGT > w1118* and *EGT > yki^act* flies. **C** Violin plots of the expression change in the different cell clusters of genes involved in various renal functions in *EGT > w1118* and *EGT > yki^act* flies. Box plots denote the medians, means, and the interquartile range. The whiskers of each box plot are the lowest and highest expression levels. **D** qPCR analysis indicating the changes in gene expression levels in the MT of *EGT > w1118* and *EGT > yki^act* flies, $n = 3$ biologically independent experiments with 2 or 3 technical replicates. Transgene expression was induced for 8 days. Data are presented as means ± SEM. $*p < 0.05$, $**p < 0.01$, $***p < 0.001$ with student t-test.

Yki flies (Fig. 3B–D). Importantly, kidney stones were observed when *Pvr^act* was expressed in MTs (Fig. 3E). In addition, elevated uric acid levels were not observed when PDGF/VEGF was activated in SCs (Supplementary Fig. 4E), suggesting that SCs are not involved in renal dysfunction in Yki flies. Finally, neither overexpression of Pvf1 in PCs nor SCs increased uric acid levels, possibly due to a low amount of Pvf1 secreted from these MT cells (Supplementary Fig. 4E) compared to the situation with Yki gut tumors. Altogether, these observations

indicate that activating the PDGF/VEGF pathway specifically in PCs is sufficient to cause renal dysfunction and phenocopy defects observed in Yki flies.

Next, we examined the expression of the genes found to be misregulated in Yki flies, in the MT samples with PC-specific activation of PDGF/VEGF signaling (Fig. 3F, Supplementary Fig. 4F). Changes in expression of these genes were similar to those observed in Yki flies (Figs. 2D and 3F, Supplementary Fig. 4F). For instance, cell junction

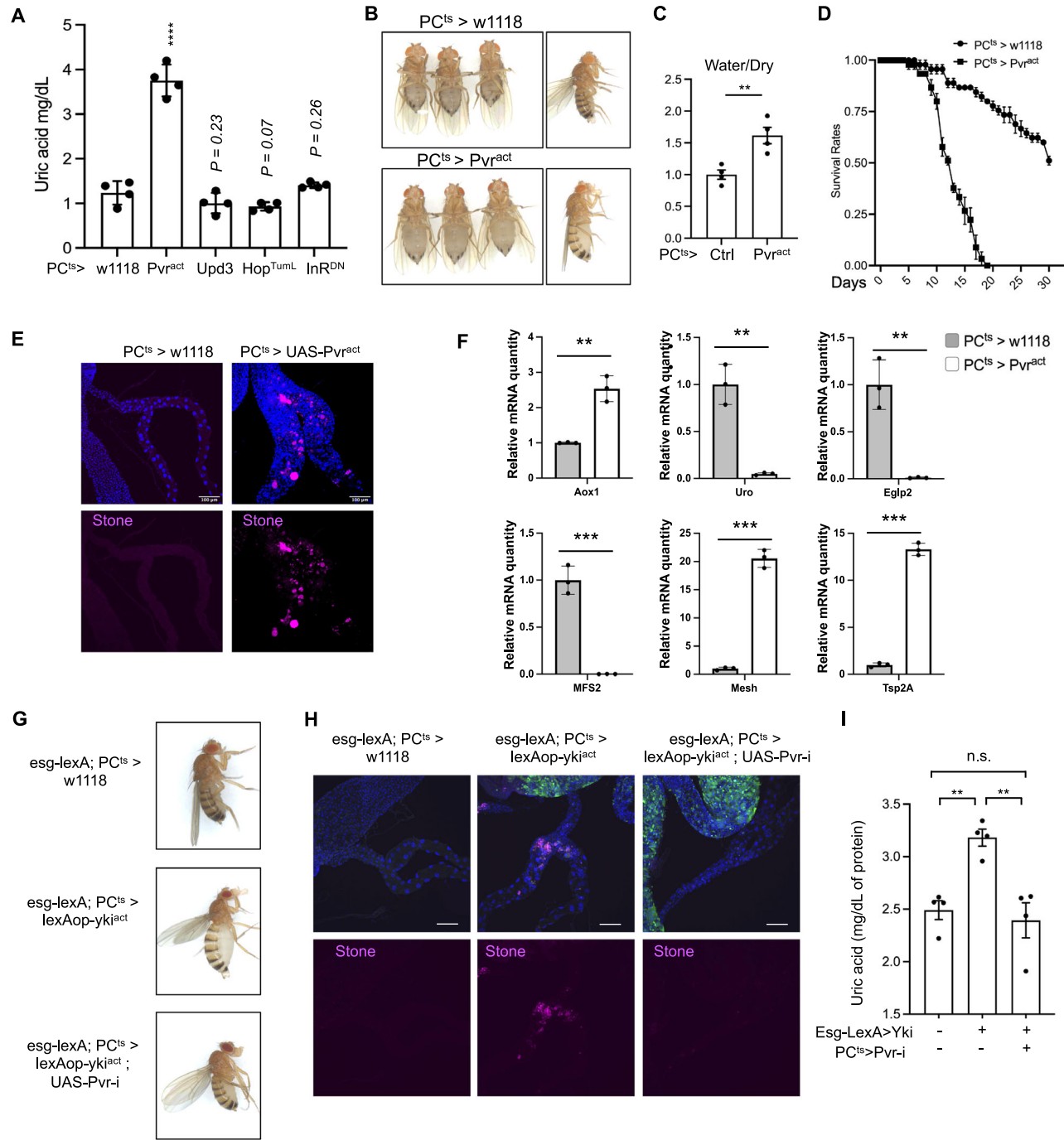

**Fig. 3 | Activation of PDGF/VEGF signaling in principal cells leads to MT dysfunction. A** Whole-body UA level in *PC^ts > Pvr^act, Upd3, Hop^TumL*, and *InR^DN* flies (*n* = 4) and *PC^ts > w1118* as control (*PC^ts* corresponds to *tub-GAL80^ts, CG31272-GAL4*). **B** Bloating phenotype associated with expression of *Pvr^act* driven by PC^ts in principal cells. **C** Ratio of fly water/dry mass. **D** Lifespan of *PC^ts > Pvr^act* and *PC^ts > w1118* control flies. *N* = 45. **E** Kidney stone images: blue is for DAPI staining to detect nuclei and purple show the kidney stones. Experiment was repeated independently five times. **F** qPCR results showing changes in gene expression levels in the MT tubule of *PC^ts > Pvr^act* and *PC^ts > w1118* flies, *n* = 3 biologically independent experiments with 2 or 3 technical replicates. **G–I** Rescue of the bloating, kidney stone and uric acid phenotypes following knockdown of *Pvr* in principal cells in the *EGT > yki^act* flies, *n* = 4 biologically independent experiments. Data are presented as means ± SEM. **p* < 0.05, ***p* < 0.01, ****p* < 0.001, *****p* < 0.0001. n.s. means no significance with student t-test.

genes, including *Inx7* (3.62, *p* = 0.0006), *Inx2* (3.46, *p* = 0.0003), *Tsp2A* (13.29, *p* = 0.0000068), *mesh* (20.55, *p* = 0.000031), *Ssk* (5.07, *p* = 0.00032) and *dlg1* (2.66, *p* = 0.013) were up-regulated (Fig. 3F, Supplementary Fig. 4E). Further, genes associated with kidney stones, including *CG31674* (0.068, *p* = 0.00013), *Vha55* (0.25, *p* = 0.012), *MFS2* (0.0021, *p* = 0.00033), and *Prestin* (0.36, *p* = 0.011), were down-regulated, and the Aquaporin genes *Eglp2* (0.011, *p* = 0.0029) and *Eglp4*

(0.0035, *p* = 0.00054) were also down-regulated (Fig. 3F, Supplementary Fig. 4E). Uric acid production-related genes *Uro* (0.048, *p* = 0.00035) and *CG30016* (0.0073, *p* = 0.000023) were significantly down-regulated, while *AOX1* (2.54, *p* = 0.0019) was up-regulated (Fig. 3F, Supplementary Fig. 4E). To further confirm that PVR signaling in MTs of Yki flies is required for the activation of these genes, we used the LexA-LexOp system to induce *yki^act* tumor in the gut and the

Gal4/UAS system to knockdown *Pvr* in PCs. Inhibition of PDGF/VEGF signaling following expression of *Pvr-i* in PCs rescued the bloating, renal stone, and uric acid phenotypes observed in tumor-bearing flies (Fig. 3G–I). Altogether, these results indicate that elevated Pvf1/PDGF/VEGF signaling in tumor flies is responsible for renal dysfunction.

## PDGF/VEGF signaling activates the JNK pathway in PCs

Pvr signaling is known to activate the Ras/Raf/MAP kinase (ERK) and JNK pathways[33]. This led us to test the roles of the ERK and JNK pathways in the MTs of Yki flies. The JNK pathway regulates the two downstream transcription factors (TFs), *kayak* (*kay*) and *Jun-related antigen* (*Jra*), both at the transcriptional and phosphorylation levels[34–37]. Interestingly, expression of both *kay* and *Jra* was upregulated in Yki flies (Fig. 4A, B, E), whereas no change in expression was observed for *anterior open* (*aop*), a TF downstream of ERK signaling (Fig. 4C). Supporting a role for JNK signaling in the MTs of Yki flies, the JNK cascade reporter gene *puckered* (*puc*) was up-regulated in the main segment PCs (Fig. 4D), an observation we confirmed by qRT-PCR (Fig. 4E). In addition, expression of *kay*, *Jra*, and *puc* was also up-regulated in MTs following PC-specific activation of the PDGF/VEGF pathway (Fig. 4F). These results indicate increased JNK pathway activity in MTs of both *Pvr*[act] and Yki flies. Consistent with these results, we observed an increase in MT JNK phosphorylation (pJNK) in both conditions at an early time point (5 days of tumor induction) and at a late time point (8 days of tumor induction) (Fig. 4G, H). Notably, the increase of JNK signaling is specific in the main segment region of the MT, since elevated levels of pJNK was only detected in the SC and PC region and not in the renal stem cell zone (SCZ) (Fig. 4I). Conversely, phosphorylation of ERK was not changed at late time points in Yki flies nor following PC-specific activation of PDGF/VEGF signaling (Supplementary Fig. 5A, B). Finally, knockdown of *Pvr* in PCs rescued elevated levels of pJNK (Fig. 4J).

Following activation of the receptor tyrosine kinase PVR by Pvf1, signaling is mediated by JNK via Crk (adapter protein that interacts with mbc), mbc (an unconventional bipartite GEF), and the kinase hep that activates bsk. Accordingly, depletion of *Crk*, *mbc*, and *hep* in the background of PVR activation flies rescued the bloating phenotype and abnormal uric acid levels (Supplementary Fig. 6A, C, D), which also partially inhibited kidney stones (Supplementary Fig. 6B) and improved the survival of these flies (Supplementary Fig. 6E). Importantly, the expression of renal genes, such as *Uro*, *CG30016*, *CapaR* and *irk2*, was restored close to normal levels (Supplementary Fig. 6F), suggesting that the JNK pathway is activated by PVR signaling through Crk/mbc/hep (Supplementary Fig. 6). Altogether, our data indicates that the PDGF/VEGF signaling activates downstream JNK pathway in the MTs of Yki flies.

## Function of PCs depends on the transcription factor *Jra*

Next, we tested whether JNK signaling plays a role in tumor-induced renal dysfunction. While knockdown of *kay* in PCs did not rescue the MT defects in Yki flies, depletion of *Jra* was able to inhibit bloating, kidney stone formation and uric acid elevation, without affecting the gut tumor (Fig. 5A–C). Importantly, inhibition of JNK signaling (by blocking *Bsk* via overexpression of *Bsk*[DN]) in *Pvr*[act] flies rescued bloating, lifespan, kidney stone formation, uric acid levels, and expression levels of genes involved in kidney function (Fig. 5D–I, Supplementary Fig. 7). Previously, we reported that *Uro* is a target of *Jra*[26], leading us to analyze the expression of *Uro* in PCs. *Uro* mRNA levels were up-regulated in MTs upon knockdown of *Jra* in PCs (Fig. 5J), suggesting that *Jra* inhibits *Uro* expression. Lower levels of *Uro* leads to insufficient oxidation of uric acid to 5-hydroxyisourate, which may contribute to the uric acid accumulation observed in Yki flies and in PCs with activated PDGF/VEGF signaling. To confirm this, we overexpressed *Uro* in PCs expressing *Pvr*[act], which rescued whole-body levels of uric acid in

these flies (Fig. 5K). Collectively, these observations indicate that PDGF/VEGF signaling regulates PCs function through the TF *Jra*.

## Pvf1 from Yki tumors activates PVR signaling in renal PCs

Yki tumors secrete the Pvf1 ligand, activating PVR signaling pathway in peripheral tissues[15]. Pvf1 shows minor expression in RSCs and is not significantly changed in RSCs in Yki flies compared to control flies (Supplementary Fig. 8). To determine whether tumor-derived Pvf1 activates PVR signaling pathway in MTs, we decreased *Pvf1* expression in the gut stem cells of Yki flies (*esg> yki*[act] + *Pvf1-i*). As reported previously, inhibition of *Pvf1* in gut tumors rescues both bloating and the water/dry mass phenotype in Yki flies without affecting the tumor (Fig. 6A, B)[15]. Strikingly, *Uro* expression and uric acid levels were decreased, and no kidney stones were observed in MTs (Fig. 6C–E). Importantly, inhibition of *Pvf1* in gut tumors rescued the expression of mis-regulated genes in the MTs of Yki flies (Fig. 6E, Supplementary Fig. 9). Collectively, our data suggest that Pvf1 from gut tumor cells remotely activates PvR signaling in MTs, leading to renal dysregulation (Fig. 6F).

## Yki tumors hijack paracrine renal PDGF/VEGF signaling

In wildtype MTs, *Pvf1* is expressed in stellate cells (SCs), a small group of cells involved in fluid secretion, and *Pvr* is widely expressed in all PCs[17,26], suggesting that Pvf1 acts as a paracrine signal from SCs to PCs. The finding that Pvf1 derived from gut tumors acts as an endocrine hormone that activates PDGF/VEGF signaling in PCs suggests that tumor secreted Pvf1 may interfere with Pvf1 paracrine signaling in MTs. To characterize and compare the physiological roles of PDGF/VEGF signaling in wildtype and Yki flies, we performed snRNA-seq analysis of MTs of control flies (*PC*[ts] > *w1118*) and flies with PDGF/VEGF signaling inhibition or activation in PCs (*PC*[ts] > *Pvr*[RN] and *PC*[ts] > *Pvr*[act], respectively) (Fig. 7A). Inhibition of PDGF/VEGF signaling in PCs did not change the cell cluster composition of MTs (Fig. 7A). However, reduction of *Pvr* activity in PCs was associated with a wide range of elevated expression of ribosomal protein genes (Supplementary Data 1), suggesting impaired ribosome biogenesis[38]. In addition, expression of many transporters including *CG7720*, *salt*, *Irk1*, *Zip48C*, and *Vha14-1* (Fig. 7B–F) was up-regulated (Supplementary Data 2), suggesting that PDGF/VEGF signaling inhibits their expression in wildtype MTs. Despite these changes in gene expression, flies with *Pvr* depletion in PCs did not show any obvious phenotypes.

Unlike reduction of PDGF/VEGF signaling, activation of this pathway in PCs led to distinct MT cell clusters (referred to hereafter as *Pvr*[act] clusters) (Fig. 7A). In total, five *Pvr*[act] clusters were identified, including two renal stem cell-like clusters, two lower segment PC-like clusters, and a cluster related to lower tubule PCs (Fig. 7A). Presumably due to the high recovery rate of cells in *Pvr*[act] clusters, other MT cell clusters, such as SCs, were not detected (Fig. 7A). Considering that SCs represent a small cell population in WT, we decided to compare gene expression changes at the whole MT level among all three samples (*PC*[ts] > *w1118*, *PC*[ts] > *Pvr*[RNAi], and *PC*[ts] > *Pvr*[act]) (Supplementary Data 2). Interestingly, 23 genes down-regulated in *Pvr*[act] MTs (Fig. 7G) were among the genes significantly up-regulated in *Pvr*[RNAi] MTs (Supplementary Data 2), suggesting that these genes are downstream targets of the PDGF/VEGF pathway. Notably, among them are 9 transporter genes: *CG7720*, *CG15406*, *CG10226*, *Irk1*, *Zip48C*, *salt*, *CG15408*, *CG6125*, and *CG4928* (Fig. 7G, Supplementary Data 2). In addition, 19 genes significantly down-regulated in *Pvr*[RNAi] MTs are up-regulated in *Pvr*[act] MTs, although no enrichment of gene groups were observed (Supplementary Data 2). These data suggest that in wildtype MTs, SCs secrete Pvf1 to down-regulate the expression of transporter genes in PCs, possibly to regulate fluid secretion and ionic balance[17].

Next, we compared the expression of the mis-regulated genes identified in the MTs of Yki flies with those of *PC*[ts] > *w1118*, *PC*[ts] > *Pvr*[RNAi],

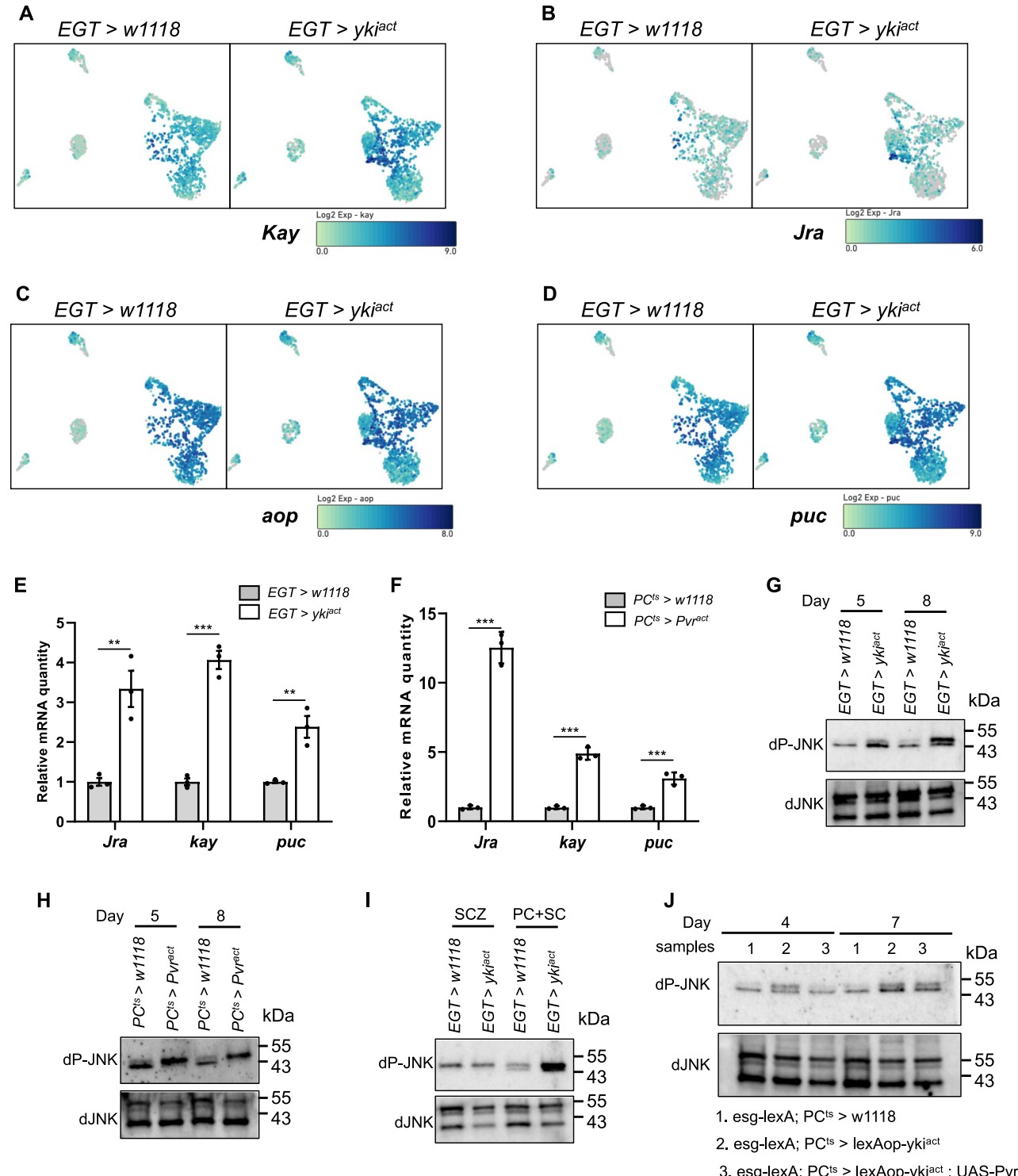

**Fig. 4 | Increase in JNK/Jra signaling in principal cells in *EGT > yki^{act}* and *PC^{ts} > Pvr^{act}* flies. A–D** UMAP showing the gene expression changes of JNK/Jra pathway genes in MT of *EGT > w1118* and *EGT > yki^{act}* flies. **E, F** qPCR of *Jra*, *kay* and *puc* genes in MT of *EGT > yki^{act}* and *PC^{ts} > Pvr^{act}* flies. Transgenes were induced for 8 days. Data are presented as means ± SEM. *$p < 0.05$, **$p < 0.01$, ***$p < 0.001$ with student t-test. **G–J** Western blots showing the protein levels of JNK and phosphorylated JNK in MT of *EGT > w1118* and *EGT > yki^{act}* **G** and *PC^{ts} > w1118* and *PC^{ts} > Pvr^{act}* **I** flies. **H** shows the signal in the stem cell zone (SCZ) and principal cells plus stellate cells (PC + SC). **J** shows the rescue of p-JNK upon knockdown of *Pvr* in principal cells in yki^{act} flies. Transgenes were induced for 5 and 8 days in **G**, **I**; 8 days in **H**; 4 and 7 days in **J**. $n = 3$ biologically independent experiments with 2 or 3 technical replicates in **E**, **F**.

and *PC^{ts} > Pvr^{act}* MTs. Interestingly, *Aox1*, *Mesh*, and *Tsp2A* were up-regulated and *Uro*, *Eglp2*, *MFS2* were down-regulated, in both *Pvr^{act}* MTs and the MTs of Yki flies (Figs. 7I–N, 2D, Supplementary Data 2), indicating that activation of the PDGF/VEGF pathway in PCs in wildtype flies phenocopies the activation of this pathway in Yki flies. In addition,

we compared the single-cell transcriptome of MT cells in five conditions (*PC^{ts} > w1118*, *PC^{ts} > Pvr^{RNAi}*, *PC^{ts} > Pvr^{act}*, *esg > w1118*, and *esg> yki^{act}*) (Fig. 7H). Interestingly, genes mis-regulated in the MTs of Yki flies were less changed than in MTs expressing *Pvr^{act}* in PCs of wildtype flies (Fig. 7I–N, I'–N'), most likely reflecting that the Pvr pathway in Yki flies

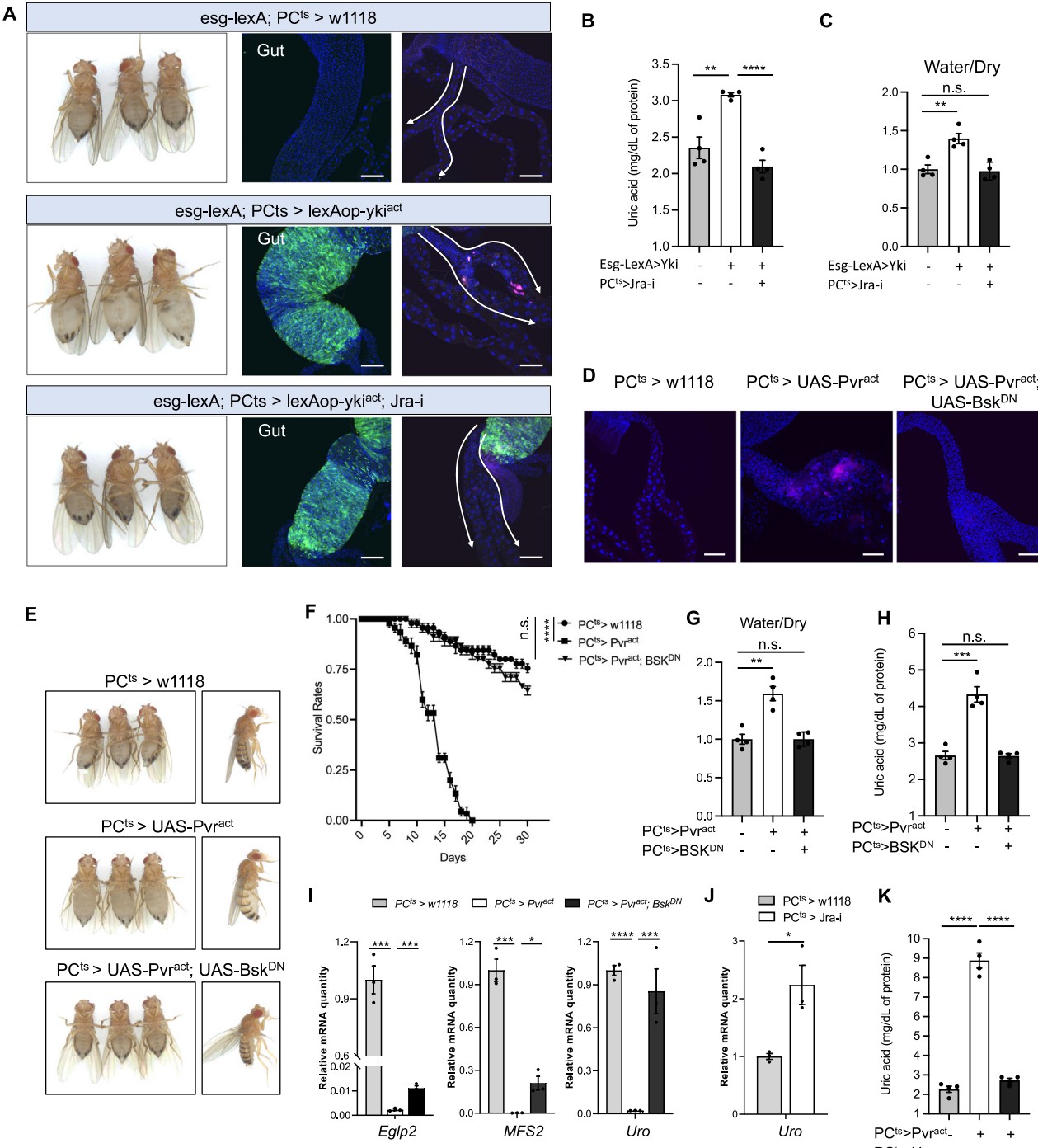

**Fig. 5 | *Jra* targets *Uro* to control uric acid metabolism in MT. A–C** Rescue of the bloating, kidney stone and uric acid phenotypes upon knocking down *Jra* in principal cells in yki[act] flies. Transgenes were induced for 6 days, $n = 4$ biologically independent experiments. Rescue of **D** kidney stones, **E** bloating, **F** survival, **G** water/dry mass, **H** UA levels, and **I** mis-regulated gene expression (*Eglp2, MFS2,* and *Uro*) when *Pvr[act]* is co-expressed with *Bsk[DN]* in principal cells. $n = 45$ for each genotype in F, $n = 4$ biologically independent experiments in **G, H,** $n = 3$ biologically independent experiments with 3 technical replicates in **I. J** qPCR of *Uro* in the MT of *PC[ts] > w1118* and *PC[ts] > Jra-i* flies. Transgenes were induced for 8 days, $n = 3$ biologically independent experiments with 3 technical replicates in **I. K** Rescue of uric acid level when *Pvr[act]* is co-expressed with *Uro* in principal cells, $n = 4$ biologically independent experiments. Data are presented as means ± SEM. *$p < 0.05$, **$p < 0.01$, ***$p < 0.001$, ****$p < 0.0001$ with student t-test.

is not as active as in *PC[ts] > Pvr[act]*. Consistent with this, MT cells of Yki flies appear to be a mixture of wildtype cells and *Pvr[act]* cells (Fig. 7H), and the mis-regulated genes identified in Yki flies displayed higher changes in *Pvr[act]* MTs (Supplementary Table 1, Supplementary Data 2). Altogether, our results suggest that elevated Pvf1 derived from Yki gut tumors hijacks paracrine renal PDGF/VEGF signaling, leading to paraneoplastic defects in the renal system (Fig. 7O).

## Discussion

Paraneoplastic syndromes in cancer patients originate from the dysfunction of organs at a distance from tumors or their metastases. Despite the clinical relevance of paraneoplastic syndromes, relatively little is known about the mechanisms underlying their pathogenesis. In this study, we demonstrate that tumor-secreted Pvf1 activates the PDGF/VEGF/JNK pathway in the PC cells of MTs, leading to mis-

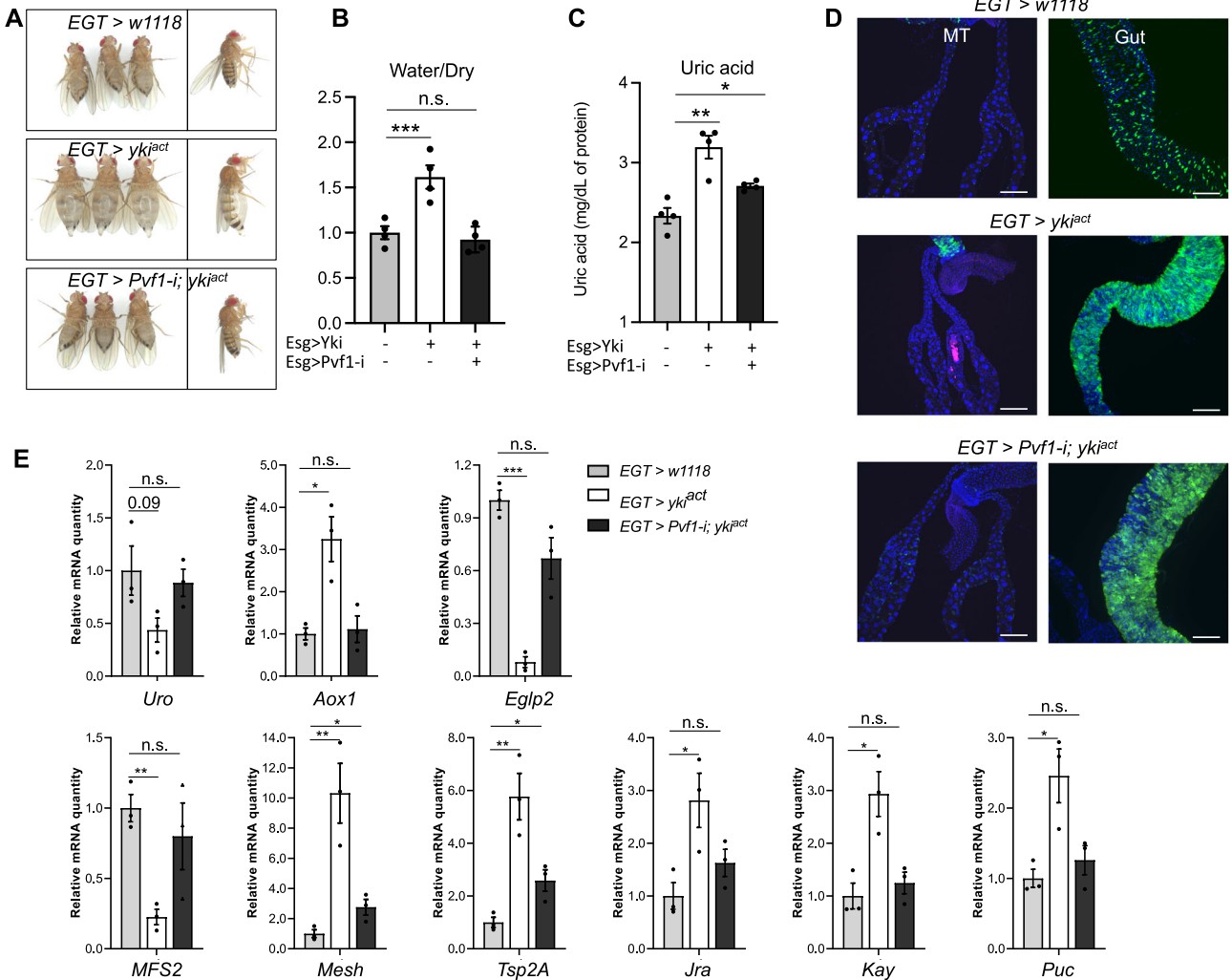

**Fig. 6 | Knockdown of *Pvf1* in *EGT > yki^act* flies repairs MT function. A–D** Rescue of the bloating, uric acid, and kidney stone phenotypes upon knocking down *Pvf1* in gut stem cells in yki^act flies. Transgenes were induced for 8 days, *n* = 4 biologically independent experiments in **B**, **C**. **E** qPCR of kidney function and JNK pathway genes in the MT of *EGT > w1118*, *EGT > yki^act* and *EGT > Pvf1-i; yki^act* flies, *n* = 3 biologically independent experiments with 3 technical replicates. Data are presented as means ± SEM. *\*p < 0.05, \*\*p < 0.01, \*\*\*p < 0.001* with student t-test.

expression of genes involved in aquaporin expression, kidney stone formation, cation transport, diuretic control, and uric acid metabolism. This pathological mis-regulation in gene expression leads to renal dysfunction and reduced viability, demonstrating a molecular mechanism of tumor-induced renal dysfunction.

Tumors hijack host signaling pathways to induce paraneoplastic syndromes. In the case of Yki flies, gut tumor-secreted Pvf1 targets muscle and adipose tissue, and activates the MEK/MAPK pathway, which induces muscle wasting and lipid loss that contribute to body wasting[15]. Our recent study suggests the elevated levels of *Pvf1* in Yki flies is due to both increased number of progenitor cells and the transcriptional activity of *Pvf1* in gut tumors[39]. Here, we demonstrate that, in addition, tumor secreted Pvf1 activates the JNK pathway in MTs, leading to renal dysfunction. Abnormal activation of PDGF/VEGF/JNK signaling in PC cells of MTs leads to up-regulation of two key renal genes, *Aox1* and *Uro*, impairing uric acid excretion[21,26,30] and resulting in an increase in uric acid levels. Further, high levels of uric acid and downregulation of *MFS2* induce formation of kidney stones[23,31]. In addition, down-regulation of aquaporins such as *Eglp2* influence water transport[40], which likely contributes to the excessive accumulation of body fluids observed in Yki flies. Interestingly, we observe a similar transcriptional up-regulation of *Aox1* and down-regulation of *Uro*,

*MFS2* and *Eglp2*, as well as similar phenotypes, i.e., elevated levels of uric acid, kidney stone formation and bloating, in both Yki flies and flies with activated PDGF/VEGF signaling in PCs. Altogether, this suggests that the renal phenotypes observed in Yki flies are the result of activation of the PDGF/VEGF pathway in PCs. Notably, the paraneoplastic phenotypes in muscle and fat, and MTs, appear to result from activation of different signal transduction cascades, ERK in the case of muscle and fat, and JNK in MTs, reflecting tissue-specific differences in response to Pvf1. In addition, bloating phenotypes have also been observed in two other fly models, an ovarian tumor model and an eye tumor model, suggesting possible renal dysfunction in these flies[41,42]. Further studies are needed to investigate the similarities or differences in renal dysfunction in these various models.

In addition to identifying the role of Pvf1 as a paraneoplastic factor, our study implicates Pvf1 as a paracrine signal in wildtype MTs. Our recent study of cell composition of wildtype MTs indicated that *Pvf1* is expressed in SCs and that *Pvr* is enriched in PCs[26]. Strikingly, inhibition of *Pvr* in MT PCs led to an increase in expression of several transporter genes, suggesting that Pvf1 from SCs regulates fluid secretion and ionic balance in PCs[17]. Previously, we showed that Pvf1 from the muscles of adult flies regulates lipid synthesis in oenocytes[43]. It will be of interest to examine whether muscle derived Pvf1 can also regulate MT physiology.

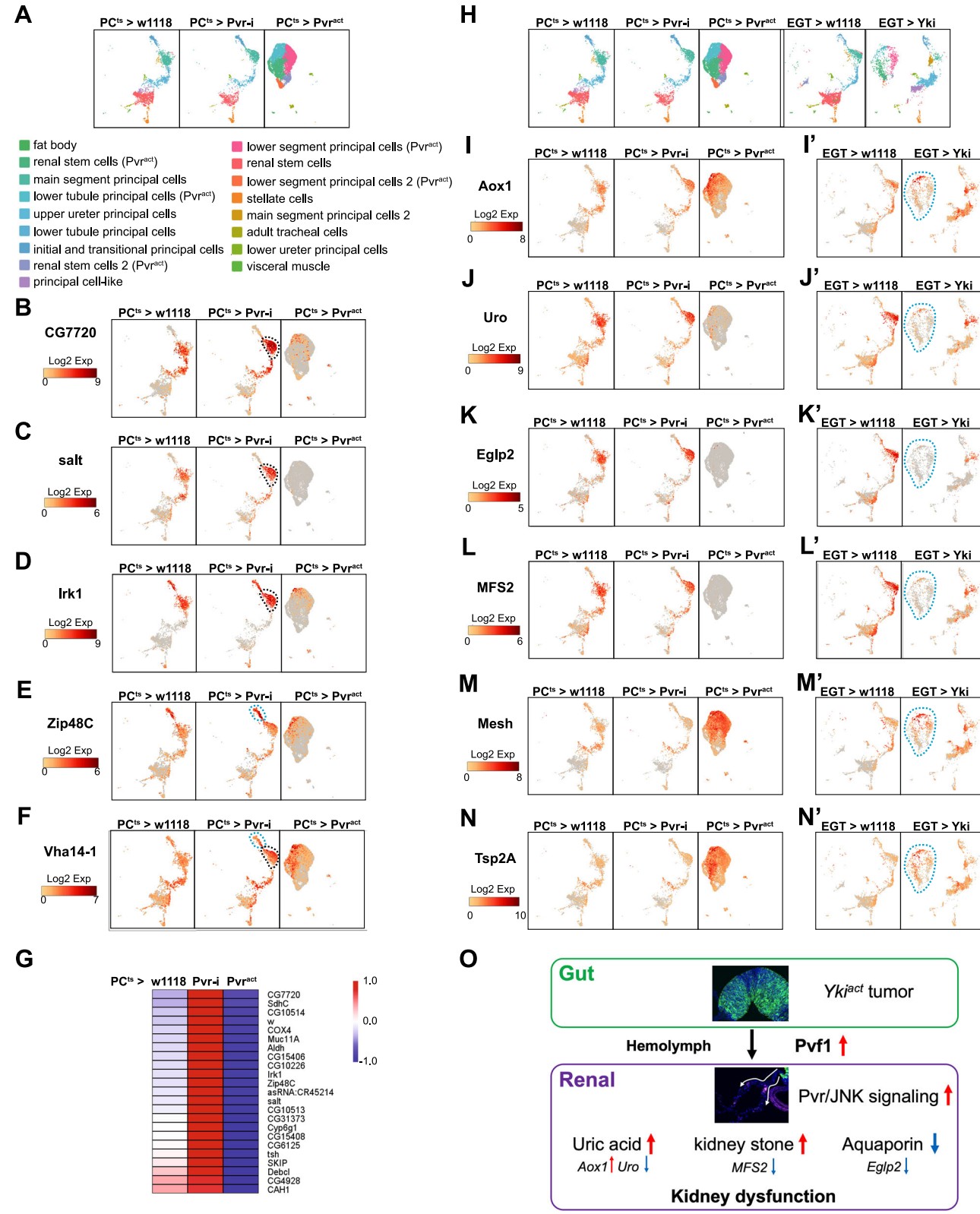

**Fig. 7 | Gene expression in the MTs of *yki^act* flies and flies with PDGF/VEGF signaling activation or inhibition. A** UMAP indicating the MT cell types of flies with PDGF/VEGF signaling activation or inhibition. UMAP indicating **B** *CG7720*, **C** *salt*, **D** *Irk1*, **E** *Zip48C* and **F** *Vha14*−1 expression levels in MTs of control and flies with PDGF/VEGF signaling activation or inhibition. Dashed black circles indicate main segment principal cells and dashed cyan circles indicate initial and transitional principal cells. **G** Heatmaps showing expression changes of Pvr downstream target

genes. **H** UMAP indicating the cell clustering of MT cells of control, Yki flies, and flies with PDGF/VEGF signaling activation or inhibition. UMAP indicating (**I&I'**) *Aox1*, (**J&J'**) *Uro*, (**K&K'**) *Eglp2*, (**L&L'**) *MFS2*, (**M&M'**) *Mesh*, and (**N&N'**) *Tsp2A* expression levels in MTs of control, Yki flies, and flies with PDGF/VEGF signaling activation or inhibition. Dashed cyan circles indicate *Pvr^act* clusters in Yki fly MTs. **O** Model of *yki^act* gut tumor induced renal dysfunction.

Interestingly, loss of autocrine PDGF/VEGF signaling in wild-type MTs has no obvious phenotypes, although the expression of a number of transporter genes suggests a role of PVR signaling in regulating fluid secretion and ionic balance[17]. By contrast, elevated Pvf1 levels in Yki flies generate severe MT dysfunction and affect fly viability. These observations suggest that the physiological/pathological role of PDGF/VEGF depends on its activation levels. In mammals, VEGF signaling is important in formation and growth of blood vessels, and PDGF functions as a growth factor, which promotes proliferation and motility of mesenchymal and other cell types[44,45]. Both PDGF and VEGF signaling are important for normal development of the kidney, as evidenced from the kidney failure and death before or at birth of PDGF beta-receptor mutant mice[46]. In addition, abnormal glomeruli lacking capillary tufts and systemic edema phenotypes are observed in newborn mice with inhibition of VEGF signaling[47]. Interestingly, in mice and humans, PDGF and VEGF signaling mediate paracrine interactions among different types of kidney cells[48,49]. In addition, increased activity of PDGF and VEGF signaling leads to kidney diseases such as renal fibrosis[50,51]. These observations are reminiscent to the aberrant activation of paracrine PDGF/VEGF signaling leading to renal dysfunction observed in the MT of Yki flies. Further, both PDGF and VEGF signaling have reported roles in cancer progression[52,53]; however, a role for PDGF/VEGF/JNK signaling in kidney physiology and paraneoplastic renal syndrome remains to be explored. Finally, defective kidney function has been reported in patients with cancer. While the cause of renal dysfunctions has been mainly attributed to cancer treatment, our study raises the possibility that tumor-secreted factors such as hormones and cytokines may contribute to the syndrome. Thus, it may be worth exploring whether kidney dysfunction in cancer patient involves a paraneoplastic role of PDGF/VEGF signaling.

## Methods

### Drosophila stocks

Fly husbandry and crosses were performed under a 12:12 hour light:dark photoperiod at 18 °C, 25 °C, or 29 °C, as indicated in each experiment. EGT (esg-GAL4, UAS-GFP, tub-GAL80$^{TS}$), esg-LexA and LexAop-yki$^{3SA}$-GFP are from the Perrimon lab stock collection.

The following strains were obtained from the Bloomington Drosophila Stock Center (BL), Vienna Drosophila Resource Center (V) and flyORF: tsh-GAL4; tub-GAL80$^{TS}$ (BL# 86330), CG31272-GAL4 (BL# 76171), UAS-yki$^{3SA}$ (BL# 28817), UAS-Pvr.lambda (BL# 58428), UAS-Pvr.lambda (BL# 58496), UAS-Pvr-RNAi (BL# 37520), UAS-upd3 is a kind gift from Dr. Frederic Geissmann, UAS-InR.DN (BL# 8253), UAS-Jra-RNAi (BL# 31595), UAS-Pvf1-RNAi (V#102699), and UAS-Pvf1-3xHA (#F002862). UAS-Uro-127D01 fly containing the 127D01 nanotag (Xu et al., 2022) was generated in this study. Uro ORF was cloned from the pENTR-Uro (DGRC, #1656531) using the primers: ACTCTGAATAGGGAATTGGGAATTCCAAAATGTTTGCCACGCCCCTCAG & CAGATCAGAACTAGTTTGCTCTAGATTAATCCTCGCCTTTCCAGAAATCTTCAAAACTTCCTGAACCCAGGTGACTATTGATGTTCT, inserted into pWalium10 vector (DGRC, #1470). The vector was injected into attP line y,w; P{nos- phi-C31\int.NLS}X; P{CaryP}attP2 to generate the transgenic flies. Female flies are used in the analysis as they showed more significant and consistent bloating phenotype.

### Drosophila culture and drug treatment

Flies were raised on standard lab food (corn meal/agar medium). For conditional expression using tubGal80$^{ts}$, flies were grown at 18 °C until eclosion, maintained at 18 °C for an additional 2d, and then shifted to 29 °C to induce expression. The concentration of chemicals mixed in the fly food was based on previously published reports[21,23]. The Super CitriMax Garcinia Cambogia extract (Swanson) contains 60% hydroxycitric acid and was purchased from Amazon. Fly food was melted and mixed with a final concentration of 30 mg/ml Garcinia Cambogia extract[23]. For NaOx feeding, a final concentration of 5 mg/ml sodium oxalate was mixed into melted fly food[23]. The high purine food contains 20 mM adenine (Sigma-Aldrich, # A8626) and 20 mM guanine (Sigma-Aldrich, # G11950)[21].

### Lifespan analysis

For survival analysis, flies were collected within 24 h of eclosion (20 female flies and 5 males per vial) and cultured on normal lab fly food at 18 °C and 60% humidity with 12 h on/off light cycle. After two days, matured and mated female flies were shifted to 29 °C to induce gene expression. Flies were transferred to fresh vials every other day to keep the vials clean, and dead flies were counted every day.

### Uric acid measurement

Uric acid was measured using the QuantiChrom$^{TM}$ Uric Acid Assay Kit (Bioassay, DIUA-250). Whole fly carbohydrates and triglycerides were measured as described previously[15,19]. To prepare fly lysates for metabolic assays, we homogenized 4 flies from each group with 300 µl PBS supplemented with 0.2% Triton X-100 and heated at 70 °C for 10 min. The supernatant was collected after centrifugation at 3000 g for 1 min at 4 °C. 10 µl of supernatant was used for protein quantification using Bradford Reagent (Sigma, B6916-500ML). Whole-body uric acid levels were measured from 5 µl of supernatant at 22 °C for 30 min using uric acid assay kit following the manufacturer's protocol.

### Single nucleus isolation and sequencing

90 pairs of Drosophila MTs were dissociated for each sample and single nuclei were prepared as previously described[26] with a few modifications. MT were dissected under a microscope from EGT > w1118 and EGT > yki$^{act}$ female adult flies after incubation at 29 °C for 8 days. Ten flies at a time were dissected and samples immediately transferred into 1.7 ml EP tube with Schneider's medium on ice to avoid exposing the tissues to room temperature for a long period of time. Once 30 flies were dissected, EP tubes were sealed with parafilm and transferred into liquid nitrogen to quickly freeze the sample and storing samples in 100 µl Schneider's medium at −80 °C for long-term. After thawing, samples were spined down using a bench top spinner, medium was discarded, and 1000 ul homogenization butter was added. Nuclei were released by homogenizing a sample on ice with a 1 ml dounce homogenizer. The homogenized sample was then filtered through a cell strainer (35 µm) and 40 µm Flowmi. After centrifugation for 10 min at 1000 g at 4 °C, nuclei were resuspended in PBS/0.5% BSA with RNase inhibitor. Before FACS sorting, samples were filtered again using 40 um Flowmi and nuclei were stained with Hoechst 33342. Ten thousand nuclei per sample were collected by FACS and loaded into a Chromium Controller (10× Genomics, PN-120223) on a Chromium Single Cell B Chip (10× Genomics, PN-120262), and processed to generate single cell gel beads in emulsion (GEM) according to the manufacturer's protocol (10× Genomics, CG000183). The library was generated using the Chromium Single Cell 3′ Reagent Kits v3.1 (10× Genomics, PN-1000121) and Chromium i7 Multiplex Kit (10× Genomics, PN-120262) according to the manufacturer's manual. Quality control for the constructed libraries was performed by Agilent Bioanalyzer High Sensitivity DNA kit (Agilent Technologies, 5067-4626). Quantification analysis was performed by Illumina Library Quantification Kit (KAPA Biosystems, KK4824). The library was sequenced on an Illumina NovaSeq system.

### snRNAseq dataset processing

The count matrices for each sample were generated using Cellranger count 7.0.0 under default setting and then imported into Seurat (4.3.0). All count data were normalized and scaled before batch corrected using Harmony (0.1.0) and subsequently reduced to 2-D Uniform Manifold Approximation and Projection (UMAP). With the

corrected Harmony embeddings, each cell was connected with its closest 20 neighbors and assigned to clusters by Louvain algorithm under Resolution=0.4. Differentially expressed genes for each cluster were computed with Wilcox Rank Sum test.

## Quantitative RT-PCR

Total RNA was extracted from 30 MT pairs per genotype from female flies per genotype using NucleoSpin RNA kit (Fisher Scientific, # NC9581114). cDNAs were synthesized using the iScript cDNA synthesis kit (Bio-Rad, #1708896) in a 20 µl reaction mixture containing 500 ng total RNA. Quantitative real-time RT-PCR (RT-qPCR) assays were performed using iQ SYBR Green Supermix (Bio-Rad, #1708880) on a CFX96 Real-Time PCR Detection System (Bio-Rad). RT-qPCR reactions were carried out with gene-specific primers (Supplementary Data 3). A 5-fold serial dilution of pooled cDNA was used as the template for standard curves. Quantitative mRNA measurements were performed in three independent biological replicates and three mechanical replicates, and data were normalized to the amount of *Dmrp49* mRNA.

## Immunostaining and confocal microscopy

*Drosophila* MTs connected to guts were dissected from adult females and fixed in 4% paraformaldehyde in phosphate-buffered saline (PBS) at room temperature for 1 hour, incubated for 1 h in Blocking Buffer (5% normal donkey serum, 0.3% Triton X-100, 0.1% bovine serum albumin (BSA) in PBS), and stained with primary antibodies overnight at 4 °C in PBST (0.3% Triton X-100, 0.1% BSA in PBS). Primary antibodies were mouse anti-GFP (Invitrogen, A11120; 1:300) and mouse anti-discs-large (DSHB, 4F3,1:50). After primary antibody incubation, the tissues were washed 4 times with PBST, stained with 4′,6-diamidino-2-phenylindole (DAPI) (1:2000 dilution) and Alexa Fluor-conjugated donkey-anti-mouse (Molecular Probes, 1:1000), in PBST at room temperature for 2 h, washed 4 times with PBST, and mounted in Vectashield medium. The kidney stones can be observed under the far-red channel.

All images presented in this study are confocal images captured with a Nikon Ti2 Spinning Disk confocal microscope. Z-stacks of 15-20 images covering one layer of the epithelium from the apical to the basal side were obtained, adjusted, and assembled using NIH Fiji (ImageJ), and shown as a maximum projection.

## Western blots

20 MTs pairs per genotype from female flies were dissected in PBS, placed in 30 µl 2xSDS sample buffer (Thermo Scientific, #39001) containing 5% 2-Mercaptoethanol at 100 °C for 10 minutes, ran 4ul on a 4%-20% polyacrylamide gel (Bio-Rad, #4561096), and transferred to an Immobilon-P polyvinylidene fluoride (PVDF) membrane (Millipore, IPVH00010). Membranes were blocked by 5% skim milk in 1x Tris-buffered saline (TBS) containing 0.1% Tween-20 (TBST) at room temperature for 30 minutes. The following primary antibodies were used: mouse anti-tubulin (Sigma, T5168, 1:10,000), rabbit anti-JNK Antibody (D-2) (Santa Cruz, sc-7345, 1:1000), rabbit phospho-JNK (Cell Signaling, 4668 T, 1:1000), rabbit anti-ERK Antibody (Cell Signaling, 4695, 1:1000), rabbit phospho-ERK (Cell Signaling, 4370, 1:1000). After washing with TBST, signals were detected with enhanced chemiluminescence (ECL) reagents (Amersham, RPN2209; Pierce, #34095). Western blot images were acquired by Bio-Rad ChemiDoc MP.

## Fly weights and bloating measurements

To measure wet and dry weights, four flies were placed into a 1.5 mL measured centrifuge tube (record as a), and wet weight was measured on Sartorius BCE64i-1CCN (record as b). Flies were then dried at 65 °C for 4 h in the drying oven with the centrifuge tube opened. Dry weight was then measured (record as c). Then, dried flies were discarded and the weight of the empty centrifuge tube was measured (record as d). Water weight was: b-a. Dry weight was: c, d.

Bloating rates measurement have been previously described[15,19]. We used three vials for each genotype with each vial contains 10 female flies. Flies were moved to 29 degree for 8 days then observed under optical microscope to measure the bloating phenotype[19].

## Data analysis, statistics, and reproducibility

All quantitative data presented in the figures were analyzed using student two-tailed t-tests in GraphPad Prism 9 software (GraphPad Software, San Diego, CA, USA). For immunostainings, experiment was repeated independently three times and one representative image was shown. No data was excluded from analysis. No randomization or blinding was done during experiments and data analysis. No statistical method was used to predetermine sample size. Statistical analysis on the quantification of protein and uric acid content in whole body was done on pooled results from two independent experiments.

## Reporting summary

Further information on research design is available in the Nature Portfolio Reporting Summary linked to this article.

## Data availability

Data generated and used in this study are described within the Article, Supplementary Data files, Supplementary Information, and Source Data file. Raw snRNA-seq reads have been deposited in the NCBI Gene Expression Omnibus (GEO) database under accession code GSE251715. Processed datasets can be mined through a web-tool [https://www.flyrnai.org/scRNA/kidney_Pvr_Yki/] that allows users to explore genes and cell types of interest. Source data are provided in this paper.

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

## Acknowledgements

We thank the assistance provided by the Microscopy Resources on the North Quad (MicRoN) core and Dr. Wenjuan Cai at CEMPS, and Biopolymers Facility at Harvard Medical School. We thank Dr. Frederic Geissmann for sharing fly stocks, Mikhail Kouzminov for help with snRNA-sequencing data analysis, and Stephanie Mohr for comments on the manuscript. J.X. is supported by start-up funding from the Shanghai Institute of Plant Physiology and Ecology/Center for Excellence in Molecular Plant Sciences, Chinese Academy of Sciences. Y.L. is

supported by Sigrid Jusélius Foundation and the Finnish Cultural Foundation (Suomen Kulttuurirahasto). N.P. is supported in part by the Cancer Grand Challenges partnership funded by Cancer Research UK (CGCATF-2021/100022) and the National Cancer Institute (1 OT2 CA278685-01). N.P. is an investigator of Howard Hughes Medical Institute. This article is subject to HHMI's Open Access to Publications policy. HHMI lab heads have previously granted a nonexclusive CC BY 4.0 license to the public and a sublicensable license to HHMI in their research articles. Pursuant to those licenses, the author-accepted manuscript of this article can be made freely available under a CC BY 4.0 license immediately upon publication.

## Author contributions

N.P., Y.L., and J.X. conceptualized and designed the experiments. J.X. and Y.L. performed most of the experiments. F.Y.Y., Y.R.C., S.Z., and J.S.S.L. performed a subset of fly work. Weihang Chen, Aram Comjean, and Yanhui Hu performed the bioinformatics analyses. J.X., Y.L., and N.P. analyzed the data. J.X. wrote the first draft of the paper. Y.L., J.S.S.L., and N.P. edited the paper. All authors discussed the results and commented on the paper.

## Competing interests

The authors declare no competing interests.
