## [Peer Review File · Nature Communications]

REVIEWER COMMENTS

Reviewer #1 (Remarks to the Author):

Paraneoplastic syndromes occur in cancer patients and originate from dysfunction of organs at a distance from the tumor or its metastasis. However, the pathological mechanisms by which tumors influence host organs are poorly understood. In this study, Xu et al in Perrimon's lab found that Pvf1, a PDGF/VEGF signaling ligand, secreted by *Drosophila* ykiact gut tumors activates the Pvr/JNK/Jra signaling pathway in the principal cells of the kidney, leading to mis-expression of renal genes and paraneoplastic renal syndrome-like phenotypes.

Drosophila has been a ideal model organism to study organ-organ and tumor-organ communications. In this study Flies with gut tumors induced by activated yorkie (*esg*> ykiact ; referred to hereafter as Yki flies) exhibit a 'bloating phenotype' characterized by an enlarged, fluid-filled abdomen . Previous studies using the same Yki flies found similar phenotypes and that tumors secrete Pvf1 and Upd3, which activate the Pvr/MEK and JAK/STAT signaling pathways, respectively, and Impl2, which represses insulin/insulin-like signaling (IIS), in peripheral tissues^{1,2,3} . This study demonstrates that tumor secreted Pvf1 activates the JNK pathway in MTs, leading to renal dysfunction.

In addition, it was already known that abnormal activation of PDGF/VEGF/JNK signaling in PC cells of MTs leads to up-regulation of two key renal genes, *Aox1* and *Uro*, impairing uric acid excretion and resulting in an increase in uric acid levels. Further, high levels of uric acid and downregulation of *MFS2* induce formation of kidney stones. In addition, down-regulation of aquaporins such as *Egfp2* influence water transport , which likely contributes to the excessive accumulation of body fluids observed in Yki flies. Like the authors said that in this study "Interestingly, we observe a similar transcriptional up-regulation of *Aox1* and down-regulation of *Uro*, *MFS2* and *Egfp2*, as well as similar phenotypes, i.e., elevated levels of uric acid, kidney stone formation and bloating, in both Yki flies and flies with activated PDGF/VEGF signaling in PCs."

In conclusion, the manuscript may lack enough "Novelty" to be published on *Nat Commun*. It is unknown whether this is ykiact fly gut tumor specific or other gut tumors also have similar phenotypes. ykiact fly gut tumors mostly are only over proliferation, it is not like real tumors in other animal models. In addition, the manuscript did not provide any direct correlation with tumors in other animal or human models, particularly mice and human.

1. Song, W. et al. Tumor-Derived Ligands Trigger Tumor Growth and Host Wasting via Differential MEK Activation. *Dev. Cell* 48, 277-286.e6 (2019). 573
2. Ding, G. et al. Coordination of tumor growth and host wasting by tumor-derived Upd3. *Cell Rep.* 36, 109553 (2021)
3. Kwon, Y. et al. Systemic Organ Wasting Induced by Localized Expression of the Secreted Insulin/IGF Antagonist Impl2. *Dev. Cell* 33, 36–46 (2015)

Reviewer #2 (Remarks to the Author):

This study by Perrimon and colleagues described a drosophila model of paraneoplastic syndrome in which yki-induced gut tumors cause renal disorders. They show that PDGF- and VEGF-related factor 1 (Pvf1) secreted from the tumor cells alters the function of the principle cells in the malpighian tubules (MTs), leading to bloating and kidney stones. This is an extension of their previous work on yki-gut tumor-induced host wasting and explains the previously-observed bloating phenotype as a consequence of renal disorders. The findings are interesting and should be of interest to a broad readership. However, some conclusions are not fully-supported by data presented so far, and this need to be addressed during revision.

Major points:

1. One major issue is the source of Pvf1 that causes MT dysfunction. The authors show that pvf1 is derived from the gut tumors as pvf1 is upregulated in the yki gut tumors and pvf1 RNAi in esg+ cells can suppress the bloating phenotype (based on their previously work, Song et al 2019). However, in addition to intestinal progenitor cells, esg is also expressed in RSCs, which are physically much closer to the principle cells, the signaling receiving cells. The authors need to examine the possibility that pvf1 is upregulated in RSCs upon yki activation and determine whether it represents a major source of Pvf1 that leads to renal disorder. In addition, RSCs are known to have role in principle cell regeneration under stress conditions such as at the presence of kidney stones (Wang, et al., 2020). Therefore, the yki activation in esg+ cells could affect the function of RSCs (such as renewal, differentiation) as well and thus contribute to renal disorder.
2. Another related issue is about pvf1 expression in stellate cells, which normally forms a paracrine mechanism on principle cells, as proposed in this study. Is pvf1 expression altered in stellate cells in yki flies, as a potential relay mechanism to the renal disorder?
3. Figure 2A, the author identified a new 'main segment PCs 2' cluster in the Yki sample. What is the function of these cells and where are they located? Are they the major responders to the Pvf1 signal that causes renal dysfunction?

Minor points:

Pvf1 is normally expressed in gut progenitor cells and is upregulated in yki-induced gut tumors. Is the increase of pvf1 in the tumor simply because there are more progenitor cells, or the transcriptional activity of pvf1 is increased?

Page 8, line 223, "depletion of 222 Jra was able to reverse bloating..." the term "reverse" is inappropriate, unless jra depletion occurs after the bloating phenotype has been developed.

Reviewer #3 (Remarks to the Author):

The author utilized a fly gut tumor model to investigate the role of a tumor-released factor, pvf1, in manipulating renal paracrine signaling, leading to kidney stone formation in the malpighian tubule. The study demonstrates that over-activated Pvr expression in PC (principal cells) can replicate the kidney dysfunction phenotypes observed in Yki tumor flies. Moreover, heightened Pvr signaling triggers the JNK

pathway, and suppressing the JNK pathway in both Pvract and Yki flies can prevent the formation of kidney stones.

Overall, this manuscript offers a comprehensive analysis of the mechanisms underlying kidney dysfunction in tumor flies. It presents substantial data and generates novel insights into paraneoplastic syndromes. This findings in this study highlights the significance of the *Drosophila* model in understanding complex human diseases.

Overall, the manuscript is beautifully written, logically sound, although a similar paper has been published earlier, the information offered from this study is still relevant, offers novel insights and provides a clearer view on how tumor induced kidney defects leads to cachexia etc.

However, there are a few key points that require attention and resolution prior to accepting the manuscript.

1. To examine the effect of kidney stones, the author conducts a NaOx feeding experiment, which increases stone size and reduces the lifespan of Yki flies. Interestingly, the author demonstrates that feeding *Garcinia cambogia* extract, which dissolves the stones, can rescue the bloating phenotype. However, it remains unclear whether reducing the stone burden would extend the lifespan of Yki flies. Further investigation is needed to explore this aspect.
2. How JNK is activated by PVR signaling? the author used a reference, which should be explained in more detail and tested since they may have context dependent effect.
3. The author provides insightful data on Yki flies, including changes in the number of malpighian tubule cell clusters, alterations in cell numbers, and gene expression profiles within each cluster. It is important for the author to provide explanations for these changes and establish their connection to kidney dysfunction. Clarifying these points would enhance the understanding of the underlying mechanisms.
4. A notable observation is that the UMAP pictures of EGT>Yki in Figure 2 and Figure 7 exhibit differences. Figure 7 displays more cell clusters, such as fat body and muscle, compared to Figure 2. If all five samples in Figure 7 represent malpighian tubules (MTs), it would be valuable for the author to clarify whether the two UMAP pictures of EGT>Yki employ different analysis conditions or if there are other factors contributing to this variation.
5. The manuscript lacks information regarding the dye used for kidney stone staining. Including this detail would provide clarity on the methodology and ensure reproducibility.
6. It would be insightful to explore if kidney stones are exhibited in other fly tumor models. The author could consider discussing the potential presence of kidney stones in alternative fly tumor models, expanding the scope of the study's findings.
7. In Figure 1J, the induction of kidney stones in Yki flies results in reduced lifespan. However, it would be valuable to include data comparing the lifespan of flies with both tumors and kidney stones to flies with tumors alone (where kidney stones have been dissolved). This comparison would help elucidate the impact of kidney stones specifically.
8. The manuscript lacks information on how bloating was measured. It is important to address this by providing details on the methodology used to assess bloating in Yki flies.

9. The significance level of the difference in cell percentage between wild type and Yki flies is not mentioned. Including this information, such as p-values, in the relevant section (Page 25, line 672-673) would strengthen the statistical interpretation.

10. The manuscript does not specify the number of flies used for the bloating study (Page 7, line 192-193). It is crucial to include this information to ensure transparency and reproducibility.

11. The manuscript indicates that the 8-day time point is considered a late time point, but it is not clear why only data from these two days are presented. It would be helpful to provide an explanation for this choice and to inquire whether there is additional supplementary data available for other time points. Further information on the temporal dynamics of the observed phenomena would enhance the comprehensiveness of the study. (Page 8, line 211)

Dear editor and reviewers,

Thank you for reviewing our manuscript. The comments and suggestions greatly helped us to improve the quality of our manuscript. Below, we address the various issues raised, which resulted in substantial revisions of our paper. All the changes have been highlighted in blue for better clarity.

Reviewer #1 (Remarks to the Author):

Paraneoplastic syndromes occur in cancer patients and originate from dysfunction of organs at a distance from the tumor or its metastasis. However, the pathological mechanisms by which tumors influence host organs are poorly understood. In this study, Xu et al in Perrimon's lab found that Pvf1, a PDGF/VEGF signaling ligand, secreted by *Drosophila* ykiact gut tumors activates the Pvr/JNK/Jra signaling pathway in the principal cells of the kidney, leading to mis-expression of renal genes and paraneoplastic renal syndrome-like phenotypes.

Drosophila has been an ideal model organism to study organ-organ and tumor-organ communications. In this study flies with gut tumors induced by activated yorkie (esg>ykiact ; referred to hereafter as Yki flies) exhibit a 'bloating phenotype' characterized by an enlarged, fluid-filled abdomen. Previous studies using the same Yki flies found similar phenotypes and that tumors secrete Pvf1 and Upd3, which activate the Pvr/MEK and JAK/STAT signaling pathways, respectively, and ImpL2, which represses insulin/insulin-like signaling (IIS), in peripheral tissues^{1,2,3}. This study demonstrates that tumor secreted Pvf1 activates the JNK pathway in MTs, leading to renal dysfunction.

In addition, it was already known that abnormal activation of PDGF/VEGF/JNK signaling in PC cells of MTs leads to up-regulation of two key renal genes, Aox1 and Uro, impairing uric acid excretion and resulting in an increase in uric acid levels. Further, high levels of uric acid and downregulation of MFS2 induce formation of kidney stones. In addition, down-regulation of aquaporins such as Eglp2 influence water transport, which likely contributes to the excessive accumulation of body fluids observed in Yki flies. Like the authors said that in this study "Interestingly, we observe a similar transcriptional up-regulation of Aox1 and down-regulation of Uro, MFS2 and Eglp2, as well as similar phenotypes, i.e., elevated levels of uric acid, kidney stone formation and bloating, in both Yki flies and flies with activated PDGF/VEGF signaling in PCs."

In conclusion, the manuscript may lack enough "Novelty" to be published on Nat Commun. It is unknown whether this is ykiact fly gut tumor specific or other gut tumors also have similar phenotypes. ykiact fly gut tumors mostly are only over proliferation, it is not like real tumors in other animal models. In addition, the manuscript did not provide any direct correlation with tumors in other animal or human models, particularly mice and human.

1. Song, W. et al. Tumor-Derived Ligands Trigger Tumor Growth and Host Wasting via

Differential MEK Activation. *Dev. Cell* 48, 277-286.e6 (2019). 573

2. Ding, G. et al. Coordination of tumor growth and host wasting by tumor-derived Upd3. *Cell Rep.* 36, 109553 (2021)

3. Kwon, Y. et al. Systemic Organ Wasting Induced by Localized Expression of the Secreted Insulin/IGF Antagonist ImpL2. *Dev. Cell* 33, 36–46 (2015)

We thank the reviewer for the critical and constructive comments. We agree with the reviewer that *Aox1*, *Uro*, *MFS2*, and *Eglp2* are important genes for kidney function, as demonstrated by previous studies showing that mis-expression of these genes impairs renal functions. As such, it is important to characterize the role of these genes in both normal and pathogenetic conditions. Our study is the first one to demonstrate that the PDGF/VEGF/JNK signaling pathway regulates the expression of these genes, and the first one to connect abnormal activation of PDGF/VEGF signaling with renal dysfunction. Our study describes a novel endocrine role of this pathway on kidney function, which contributes to our understanding of the pathogenic role of PDGF and VEGF signaling in cancer and paraneoplastic syndromes. Importantly, the bloating phenotype was also observed in two other fly cancer models, whereby tumors were induced in the ovary and the eye (His et al., 2023; Santabárbara-Ruiz and Léopold, 2021), suggesting that kidney dysfunction occurs in various tumor models. We have added this information to the manuscript and revised the manuscript to better reflect the importance of our findings.

Hsi TC, Ong KL, Sepers JJ, Kim J, Bilder D. Systemic coagulopathy promotes host lethality in a new *Drosophila* tumor model. *Curr Biol.* 33(14):3002-3010 (2023).

Santabárbara-Ruiz P, Léopold P. An Oatp transporter-mediated steroid sink promotes tumor-induced cachexia in *Drosophila*. *Dev Cell.* 56(19):2741-2751 (2021).

Reviewer #2 (Remarks to the Author):

This study by Perrimon and colleagues described a *Drosophila* model of paraneoplastic syndrome in which *yki*-induced gut tumors cause renal disorders. They show that PDGF- and VEGF-related factor 1 (*Pvf1*) secreted from the tumor cells alters the function of the principle cells in the malpighian tubules (MTs), leading to bloating and kidney stones. This is an extension of their previous work on *yki*-gut tumor-induced host wasting and explains the previously observed bloating phenotype as a consequence of renal disorders. The findings are interesting and should be of interest to a broad readership. However, some conclusions are not fully supported by data presented so far, and this need to be addressed during revision.

We appreciate the thoughtful suggestions and encouraging comments. We have added below new data to better support our conclusions.

Major points:

1. One major issue is the source of *Pvf1* that causes MT dysfunction. The authors show that *pvf1* is derived from the gut tumors as *pvf1* is upregulated in the *yki* gut tumors and

pvf1 RNAi in *esg*⁺ cells can suppress the bloating phenotype (based on their previously work, Song et al 2019). However, in addition to intestinal progenitor cells, *esg* is also expressed in RSCs, which are physically much closer to the principle cells, the signaling receiving cells. The authors need to examine the possibility that *pvf1* is upregulated in RSCs upon *yki* activation and determine whether it represents a major source of Pvf1 that leads to renal disorder. In addition, RSCs are known to have role in principle cell regeneration under stress conditions such as at the presence of kidney stones (Wang, et al., 2020). Therefore, the *yki* activation in *esg*⁺ cells could affect the function of RSCs (such as renewal, differentiation) as well and thus contribute to renal disorder.

We thank the reviewer for the comments. We carefully analyzed *Pvf1* expression in our snRNA-seq dataset, *Pvf1* has a minor expression in RSCs and is not significantly changed in RSCs in *Yki* flies compared with control flies (Figures attached below). In addition, the overall expression level of *Pvf1* in MTs is similar between *Yki* and control flies. Thus, *Pvf1* unlikely originates from RSCs in *Yki* flies. Importantly, we observed a decrease in the number of RSCs in *Yki* flies (from 19% in control flies to 6% in *Yki* flies). Further, the expression of *Notch* (*N*) was decreased in the RSCs of *Yki* flies compared to wild type flies. In addition, genes downstream of *N*, such as *escargot* (*esg*) and *Delta* (*DI*) were also significantly decreased. Thus, the abnormal expression levels of these genes may contribute to RSCs loss in tumor flies. Altogether, we agree with the reviewer that the reduction of RSCs in *Yki* flies may contribute to renal disorder, and have now added this observation and relevant new data to the manuscript.

***Pvf1* expression in the MT cell cluster.** (A) Gene expression of *Pvf1* in the UMAP. Dotted box show the stellate cell and stem cell clusters. (B) and (C) Expression levels of *Pvf1* in *EGT > w1118* and *EGT > yki^{jact}* MTs visualized by violin plots in (B) all MT cells and (C) each cluster, respectively.

2. Another related issue is about *pvf1* expression in stellate cells, which normally forms a paracrine mechanism on principle cells, as proposed in this study. Is *pvf1* expression altered in stellate cells in *yki* flies, as a potential relay mechanism to the renal disorder?

We thank the reviewer for the suggestion. We analyzed our snRNA-seq dataset and observed an increase in *Pvf1* expression in SCs (Figures attached above), thus, we further tested role of *Pvf1/Pvr* signaling in SCs. Interestingly, overexpression of *Pvf1* in

SCs did not phenocopy renal dysfunction of Yki/Pvr PC activation flies, indicated by their normal UA levels (Figure S4E). This data suggests that *Pvf1* expression in SCs does not cause renal disorder.

3. Figure 2A, the author identified a new ‘main segment PCs 2’ cluster in the Yki sample. What is the function of these cells and where are they located? Are they the major responders to the Pvf1 signal that causes renal dysfunction?

We thank the reviewer for the comment. The “main segment principal cell 2” cluster constitutes 2% of total MT cells, whereas the “main segment principal cell” represents 23%. In addition, the dysregulated renal genes we identified in Yki flies are changed in the “main segment principal cell” cluster and across all principal cell clusters. These observations suggest the “main segment principal cell 2” is unlikely to be the major responders to the Pvf1 signal. The function of these cells does not directly connect with this study but we agree with the reviewer that this novel cell cluster is interesting and worth exploring further at a later date. In this cell cluster, the top two marker genes are *Slc45-1* and *CAHbeta*, an osmolyte/sucrose transporter and a beta-class carbonate dehydratase, respectively. We added this information.

Minor points:

Pvf1 is normally expressed in gut progenitor cells and is upregulated in yki-induced gut tumors. Is the increase of pvf1 in the tumor simply because there are more progenitor cells, or the transcriptional activity of pvf1 is increased?

We have another study currently on bioRxiv (Liu et al., 2023) indicating that the overall upregulation of *Pvf1* level in Yki flies is caused by a combination of increased progenitor cells and elevated transcriptional activity. We have added this to the discussion.

Liu Y, Dantas E, Ferrer M, Liu Y, Comjean A, Davidson EE, Hu Y, Goncalves MD, Janowitz T, Perrimon N. Tumor Cytokine-Induced Hepatic Gluconeogenesis Contributes to Cancer Cachexia: Insights from Full Body Single Nuclei Sequencing. bioRxiv. 2023 May 18:2023.05.15.540823.

Page 8, line 223, “depletion of 222 Jra was able to reverse bloating...” the term “reverse” is inappropriate, unless jra depletion occurs after the bloating phenotype has been developed.

We have revised the text accordingly.

Reviewer #3 (Remarks to the Author):

The author utilized a fly gut tumor model to investigate the role of a tumor-released factor, pvf1, in manipulating renal paracrine signaling, leading to kidney stone formation

in the malpighian tubule. The study demonstrates that over-activated Pvr expression in PC (principal cells) can replicate the kidney dysfunction phenotypes observed in Yki tumor flies. Moreover, heightened Pvr signaling triggers the JNK pathway, and suppressing the JNK pathway in both Pvract and Yki flies can prevent the formation of kidney stones. Overall, this manuscript offers a comprehensive analysis of the mechanisms underlying kidney dysfunction in tumor flies. It presents substantial data and generates novel insights into paraneoplastic syndromes. This findings in this study highlights the significance of the *Drosophila* model in understanding complex human diseases. Overall, the manuscript is beautifully written, logically sound, although a similar paper has been published earlier, the information offered from this study is still relevant, offers novel insights and provides a clearer view on how tumor induced kidney defects leads to cachexia etc. However, there are a few key points that require attention and resolution prior to accepting the manuscript.

We thank the reviewer for the encouraging comments. We have added below new data and more information accordingly.

1. To examine the effect of kidney stones, the author conducts a NaOx feeding experiment, which increases stone size and reduces the lifespan of Yki flies. Interestingly, the author demonstrates that feeding *Garcinia cambogia* extract, which dissolves the stones, can rescue the bloating phenotype. However, it remains unclear whether reducing the stone burden would extend the lifespan of Yki flies. Further investigation is needed to explore this aspect.

We thank the reviewer for the suggestion. The *Garcinia cambogia* extract contains 60% hydroxycitric acid, which inhibits the bloating phenotype by dissolving the stones. However, as a chemical not usually consumed by flies, it leads to side effects and lethality of Yki flies. Because of this, we used genetics to modify the Pvr pathway to restore renal function in Yki flies and investigated the survival benefit.

2. How JNK is activated by PVR signaling? the author used a reference, which should be explained in more detail and tested since they may have context dependent effect.

We thank the reviewer for the suggestion. We examined the role of *Crk*, *mbc*, and *hep*, that act downstream of Pvr, in regulating JNK. Our results indicate that knocking down *Crk*, *mbc*, or *hep* genes in Yki flies rescued the misexpression of renal genes, bloating phenotype, abnormal uric acid levels, and also partially rescued the kidney stone phenotype. In addition, PVR activation induced lethality could be rescued by knocking down *Crk*, *mbc*, or *hep* genes. We have added this data and more information to the manuscript accordingly.

Inhibition of PDGF/VEGF signaling components in principal cells rescues MT dysfunction associated with Pvract. (A) Bloating phenotypes associated with expression of *Pvr^{act}*, *Crk-i; Pvr^{act}*, *hep-i; Pvr^{act}* and *mbc-i; Pvr^{act}*, driven by *PC^{ts}* in principal cells. (B) Kidney stone phenotypes: blue is for DAPI staining to detect nuclei and purple indicates the kidney stones. (C) Ratio of fly water/dry mass. (D) Whole-body UA level. (E) Lifespan. N=80. (F) qPCR results showing changes in gene expression levels in the MT. Data are presented as means \pm SEM. * $p < 0.05$, ** $p < 0.01$, *** $p < 0.001$, **** $p < 0.0001$. n.s. means no significant with student t-test.

3. The author provides insightful data on Yki flies, including changes in the number of malpighian tubule cell clusters, alterations in cell numbers, and gene expression profiles within each cluster. It is important for the author to provide explanations for these

changes and establish their connection to kidney dysfunction. Clarifying these points would enhance the understanding of the underlying mechanisms.

We thank the reviewer for the helpful suggestion. We included an analysis of gene expression profiles of each cluster and explained the changes in gene expression related to kidney functions, for example in Fig.2C and D, Fig. S2. For changes in cell numbers, we added more data. Renal stem cells in tumor flies were significantly reduced from 19% to 6% of total cells (see Figures below). In line with this, *esg > GFP* positive cells in the renal stem cell zone were decreased after six days of tumor induction (see below Fig. A and B). Previous studies reported that Notch (N) signaling is involved in the kidney stem cell renewal and maintenance (see below Fig. C and D, Xu et al., 2022; Wang and Spradling, 2020). This information led us to investigate the expression of *N* in these flies. Indeed, expression of *N* was decreased in the ykiact gut tumor flies (Log2 folder change, 0.18) compared to wild type flies (0.58) (see below Fig. E). Downstream genes of N signaling, such as *escargot (esg)* and *Delta (Dl)* were also significantly decreased in ykiact gut tumor flies (see below Fig. F and G). Abnormal expression levels of these genes may contribute to the renal stem cell loss in tumor flies. The *cut (ct)* gene, which was marker of the principal cells, also showed decreased expression in the upper ureter principal cells and lower segment principal cells - these two cell clusters were located in the renal stem cell zone (see below Fig. H). Taking together, these observations suggest that renal stem cell renewal and maintenance are impaired in tumor flies. We modified the manuscript and added these results as suggested.

Xu, J. et al. Transcriptional and functional motifs defining renal function revealed by single-nucleus RNA sequencing. Proc. Natl. Acad. Sci. 119, e2203179119 (2022).

Wang C, Spradling AC. An abundant quiescent stem cell population in *Drosophila* Malpighian tubules protects principal cells from kidney stones. Elife. 16; 9: e54096 (2020).

Aberrant renal stem cells renew in flies with yki^{act} gut tumor. (A) *esg* positive renal stem cell signal decreased during the growth of *yki^{act}* gut tumor. Green indicates *esg* positive cells, blue is DAPI staining. White arrows indicate the renal stem cell zone. (B) Changes in renal stem cells number during the growth of *yki^{act}* gut tumor. Data are presented as means \pm SEM. *** $p < 0.001$, **** $p < 0.0001$. n.s. means no significant with student t-test. N=6 for each column. (C-H) Notch (N) signaling was decreased in the Malpighian tubule in *yki^{act}* gut tumor flies. C and D show results from previous studies indicating that the N pathway is activated in the renal stem cell and play a key role in the

renal stem cell renewal. E-G show the decrease in the UMAP of *N* expression and its two downstream genes, *esg* and *DI*. H show the decrease in the UMAP of *ct* expression in principal cells.

4. A notable observation is that the UMAP pictures of EGT>Yki in Figure 2 and Figure 7 exhibit differences. Figure 7 displays more cell clusters, such as fat body and muscle, compared to Figure 2. If all five samples in Figure 7 represent malpighian tubules (MTs), it would be valuable for the author to clarify whether the two UMAP pictures of EGT>Yki employ different analysis conditions or if there are other factors contributing to this variation.

We thank the reviewer for the comment. Cell numbers is the factor contributing to this variation. Fat body and muscle are “contaminants”, introduced when we dissect MTs. The cell numbers of these two tissues are very low and not enough to form individual clusters in Figure 2, in which case fat body and muscle cells are distributed to other cell clusters. For Figure 7, we combined five samples for analysis, which provided more cells of Fat body and muscle, thus allowing individual clusters to be detected. Likewise, increased numbers of cells for the combined analysis in Figure 7 enabled us to detect sub-clusters that were not clearly separated in Figure 2.

5. The manuscript lacks information regarding the dye used for kidney stone staining. Including this detail would provide clarity on the methodology and ensure reproducibility.

We added this information accordingly.

6. It would be insightful to explore if kidney stones are exhibited in other fly tumor models. The author could consider discussing the potential presence of kidney stones in alternative fly tumor models, expanding the scope of the study's findings.

We thank the reviewer for the suggestion. Bloating phenotypes, which may be associated to impaired kidney function, have been reported in two recent papers describing a fly ovarian tumor model (His et al., 2023) and a fly eye tumor model (Santabárbara-Ruiz and Léopold, 2021). We added this information in the discussion.

Hsi TC, Ong KL, Sepers JJ, Kim J, Bilder D. Systemic coagulopathy promotes host lethality in a new *Drosophila* tumor model. *Curr Biol*. 33(14):3002-3010 (2023).

Santabárbara-Ruiz P, Léopold P. An Oatp transporter-mediated steroid sink promotes tumor-induced cachexia in *Drosophila*. *Dev Cell*. 56(19):2741-2751 (2021).

7. In Figure 1J, the induction of kidney stones in Yki flies results in reduced lifespan. However, it would be valuable to include data comparing the lifespan of flies with both tumors and kidney stones to flies with tumors alone (where kidney stones have been dissolved). This comparison would help elucidate the impact of kidney stones specifically.

We thank the reviewer for the suggestion. We tried to use the *Garcinia cambogia* extract to address the role of kidney stones on lifespan. The *Garcinia cambogia* extract contains 60% hydroxycitric acid, which inhibits the bloating phenotype by dissolving the stones. However, as a chemical not usually consumed by flies, it leads to side effects and lethality of Yki flies. We are not aware of any better chemicals. Because of this, we used genetics to modify the Pvr pathway to restore renal function in Yki flies and investigated the survival benefit.

8. The manuscript lacks information on how bloating was measured. It is important to address this by providing details on the methodology used to assess bloating in Yki flies.

Thanks for the reviewer's suggestions. The bloating measurements are based on a previous study (Kwon et al., 2015). In addition, we examined wet and dry weight. Information of both methods have been added to the manuscript.

Kwon, Y. et al. Systemic Organ Wasting Induced by Localized Expression of the Secreted Insulin/IGF Antagonist ImpL2. *Dev. Cell* 33, 36–46 (2015).

9. The significance level of the difference in cell percentage between wild type and Yki flies is not mentioned. Including this information, such as p-values, in the relevant section (Page 25, line 672-673) would strengthen the statistical interpretation.

We thank the reviewer for the suggestion. Due to the cost of the snRNA-seq, we had one sample per condition each containing 90 MTs, thus the significance calculation was not included.

10. The manuscript does not specify the number of flies used for the bloating study (Page 7, line 192-193). It is crucial to include this information to ensure transparency and reproducibility.

Thanks for the reviewer's suggestions. We added the number of flies used in the method section. In addition, we added more details on the bloating analysis as suggested.

11. The manuscript indicates that the 8-day time point is considered a late time point, but it is not clear why only data from these two days are presented. It would be helpful to provide an explanation for this choice and to inquire whether there is additional supplementary data available for other time points. Further information on the temporal dynamics of the observed phenomena would enhance the comprehensiveness of the study. (Page 8, line 211)

We thank the reviewer for the suggestion. We described the development of the bloating phenotype in Yki flies in a previous study (Kwon et al., 2015; Song et al, 2019; Ding et al., 2021). Bloating starts at Day 6 and is apparent at Day 8, thus, our study focused on this time point. We added this information accordingly.

Kwon, Y. et al. Systemic Organ Wasting Induced by Localized Expression of the Secreted Insulin/IGF Antagonist ImpL2. *Dev. Cell* 33, 36–46 (2015).

Song, W. et al. Tumor-Derived Ligands Trigger Tumor Growth and Host Wasting via Differential MEK Activation. *Dev. Cell* 48, 277-286. (2019)

Ding, G. et al. Coordination of tumor growth and host wasting by tumor-derived Upd3. *Cell Rep.* 36, 109553 (2021)

REVIEWERS' COMMENTS

Reviewer #1 (Remarks to the Author)

I read response letter and revised manuscript. I recommend it to be accepted for publication although they have not fully answered my questions,

Reviewer #2 (Remarks to the Author):

The majority of the concerns have been addressed by the authors through the implementation of new analyses and/or by offering reasonable explanations. However, I still have one remaining point that requires attention from the authors.

Could the authors provide explanations as to why the overexpression of Pvf1 in SCs does not lead to renal disorder? Is it possible that Pvf1 overexpression in SCs fails to hyper-activate Pvr signaling in PCs?

Reviewer #3 (Remarks to the Author):

In this revised version, the author addressed the reviewer's questions and provided additional data to support the central concept that tumor-secreted Pvf1 induces kidney stones through the JNK pathway in host Malpighian tubules. The experiments involving the downregulation of Crk, mbc, and hep clearly illustrate the regulatory role of JNK in kidney stone formation. Further explanation of tumor single-cell data introduces novel perspectives on the damage inflicted upon Malpighian tubules. Although the same phenotype has been previously reported, this study reveals a new mechanism that warrants further exploration.

Reviewer #2 (Remarks to the Author):

The majority of the concerns have been addressed by the authors through the implementation of new analyses and/or by offering reasonable explanations. However, I still have one remaining point that requires attention from the authors.

Could the authors provide explanations as to why the overexpression of Pvf1 in SCs does not lead to renal disorder? Is it possible that Pvf1 overexpression in SCs fails to hyper-activate Pvr signaling in PCs?

We added this sentence to the manuscript as a likely explanation: "Finally, neither overexpression of Pvf1 in PCs nor SCs increased uric acid levels, possibly due to a low amount of Pvf1 secreted from these MT cells (Fig. S4E) compared to situation with gut tumors."